# FIXING DATA AUGMENTATIONS FOR OOD DETECTION

## ABSTRACT

Out-of-distribution (OOD) detection methods, especially post-hoc methods, rely on off-the-shelf pre-trained models. Existing literature shows how OOD and ID performance are correlated, *i.e.* stronger models with better ID performance tend to perform better in OOD detection. However, significant performance discrepancies exist between model versions, sometimes exceeding the impact of the OOD detection methods themselves. In this study, we systematically investigated this issue and identified two main factors—label smoothing and mixup—that, while improving in-distribution accuracy, lead to a decline in OOD detection performance. We provide empirical and theoretical explanations for this phenomenon and propose a solution that enhances OOD Detection while maintaining strong in-distribution performance. Code will be released upon acceptance.

## 1 INTRODUCTION

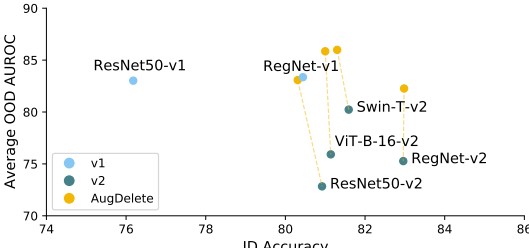

Figure 1: OOD and ID performance comparison between `torchvision` v1, v2 and AugDelete (ours) models on ImageNet-1K. AUROC is averaged among near-OOD and far-OOD datasets.

Out-of-distribution (OOD) detection identifies input samples that differ from the in-distribution (ID) training data. Detecting such samples avoids overconfident or incorrect predictions on data outside the training scope, and is particularly important in sensitive domains such as healthcare, autonomous driving, and security. Previous works (Vaze et al., 2022b; Lu et al., 2024) have shown that a model's ID and OOD detection performance are correlated - the higher the ID classification accuracy (on CIFAR, ImageNet, *etc.*), the better it is at distinguishing OOD versus ID samples. They assume that a stronger separation of ID classes naturally leads to a separation of OOD from ID classes. The generalization improvement may come from the learning rate schedule or model ensemble, though data augmentation has been found to be the most effective Lu et al. (2024). An assortment of data augmentation strategies, such as RandAugment (Cubuk et al., 2020), Style Augment (Geirhos et al., 2018), and AugMix (Hendrycks et al., 2020a) have all been found to be effective for improving both ID and OOD performance. These strategies use a combination of techniques, such as image rotation, translation, or color transformation.

Curiously, our empirical results challenge the conventional understanding of ID and OOD performance correlation. Specifically, we find two commonly used augmentation strategies – label smoothing (Szegedy et al., 2016) and mixup (Zhang, 2017) exhibit the opposite phenomena. Models trained with label smoothing and mixup have a $\sim 10\%$ drop in OOD detection performance compared to omitting these strategies, despite their ability to improve model (ID) accuracy. This phenomenon is observed across a range of convolutional and transformer network architectures, including ResNet, MobileNet, ResNetXt, WideResNet, RegNet, SWIN-T, and ViT (See Figure 1 and appendix).

This naturally begs the question - why should label smoothing and mixup harm OOD detection? These two augmentation strategies exhibit a trade-off between ID and OOD performance, contradicting trends in prior literature. One hint lies in the transformation in the augmentation itself. Label-smoothing and mixup transform the data sample's label. In contrast, previously reported augmentation strategies that improve ID and OOD separation operate only on the data sample itself, *i.e.* RandAugment, Style Augment, and AugMix, as shown by Lu et al. (2024).

We analyze, from a gradient perspective, to show that both label smoothing and mixup reduce the maximal logit values. This reduction is more pronounced for in-distribution (ID) samples than out-of-distribution (OOD) samples, thereby diminishing their separability. In turn, methods that rely on the logit values for OOD separation, such as the maximal logit score (MLS) (Hendrycks et al., 2022) or energy-based score (EBO) (Liu et al., 2020), are compromised. Feature-based methods are likely similarly compromised, due to downstream effects of diminished gradients being back-propagated from logits.

To address this issue, we propose two novel methods: Augmentation Deletion (AugDelete) for finetuning pretrained models and Augmentation Revision (AugRevise) for models trained from scratch. AugDelete mitigates the negative effects of label smoothing and mixup by deleting them and finetuning only the final layer of the network. In contrast, AugRevise introduces a revised data augmentation method paired with a corresponding training strategy aimed at enhancing in-distribution generalization while preserving OOD detection. Both AugDelete and AugRevise demonstrate improvements over baseline methods in OOD detection (see Fig. 1). Besides, AugRevise outperforms state-of-the-art training-based methods regrading OOD detection and ID accuracy.

Our contributions are as follows:

- We identified that label smoothing and mixup, two widely used data augmentation techniques used in modern neural network training, can significantly degrade OOD detection performance despite improving in-distribution accuracy.
- We theoretically demonstrated that label smoothing and mixup reduce the separation between OOD and ID data in the logit space, impairing OOD detection.
- Based on this analysis, we proposed AugDelete for finetuning pretrained models and AugRevise for training from scratch. Both methods enhance OOD detection while maintaining strong in-distribution performance.

## 2 RELATED WORKS

**Post-Hoc OOD Detection** methods often use pre-trained models; the main research focus is to define new score functions or post-hoc adjustments to improve detection capabilities. Methods such as softmax-based thresholds (Hendrycks & Gimpel, 2017a) and EBO (Liu et al., 2020) use simple yet effective techniques to repurpose off-the-shelf networks for OOD detection. Others, such as ASH (Djurisic et al., 2023) and ReAct (Sun et al., 2021), are logit-based. They modify logits by reshaping the feature activation, showing promising results in OOD detection. Feature-based methods leveraging internal representations. For instance, Mahalanobis distance-based methods (Lee et al., 2018) calculate the distance of feature vectors from class-conditional Gaussian distributions, effectively identifying OOD samples by measuring feature space uncertainty. In addition to logit-based and feature-based techniques, recent work like NNGuide (Park et al., 2023) combine the two to derive more robust OOD detection scores.

**Training-based OOD Detection** methods adjust model training to improve the model's ability to distinguish between ID and OOD samples. One strategy is through explicit supervision, either from true OOD samples (Hendrycks et al., 2019) or synthesized virtual ones (Pinto et al., 2022; Huang & Li, 2021). Synthesized samples are more appealing, since real OOD data is typically not available for training. One example is MOS (Huang & Li, 2021), which creates virtual OOD samples by grouping classes to encourage clearer separation in feature space. RegMixup (Pinto et al., 2022) treat mixed-up samples as virtual outliers, the cross-entropy loss of which serves as regularizers for strengthening the decision boundary between ID and OOD data. Other training-based techniques, like LogitNorm (Wei et al., 2022) and T2FNorm (Regmi et al., 2023), aim to improve the separability of feature representations between ID and OOD samples. After the model training, a compatible post-hoc score function is still required for OOD detection.

**Relations between ID and OOD** has been widely explored in previous works (Vaze et al., 2022b; Humblot-Renaux et al., 2024; Lu et al., 2024). Vaze et al. (2022b) found that a good closed-set classifier is able to identifying semantically novel classes; similarly, Humblot-Renaux et al. (2024) observed that ID and OOD accuracy are positively correlated, at least for correctly predicted ID samples. Lu et al. (2024) found that data augmentations including AugMix and RandAugment improve both ID and OOD performance.

## 3 PRELIMINARIES

### 3.1 OOD DETECTION

A commonly used setup for the OOD detection task is to identify semantic shifts in image classification (Huang & Li, 2021; Yang et al., 2022; Hendrycks et al., 2022). During training, only in-distribution (ID) training set $\{(\boldsymbol{x}, \boldsymbol{y}) \sim \mathcal{D}_{\text{ID}}, \boldsymbol{y} \in \mathcal{Y}_{\text{ID}}\}$ are observed, where $\mathcal{Y}_{\text{ID}}$ has $C$ ID classes. Samples from semantically novel classes unseen during training are considered OOD. During testing, OOD samples $\{(\boldsymbol{x}, \boldsymbol{y}) \sim \mathcal{D}_{\text{OOD}}, y \in \mathcal{Y}_{\text{OOD}}, \mathcal{Y}_{\text{OOD}} \cap \mathcal{Y}_{\text{ID}} = \emptyset\}$ are encountered.

To separate ID and OOD samples, a score function $S(\boldsymbol{x})$ is designed to output higher values for ID samples than OOD samples. Based on some threshold $\tau$, an OOD indicator $\mathbb{1}(\boldsymbol{x}; \tau)$ can be defined as

$$\mathbb{1}(\boldsymbol{x}; \tau) = \begin{cases} \text{ID} & \text{if } S(\boldsymbol{x}) \geq \tau, \\ \text{OOD} & \text{if } S(\boldsymbol{x}) < \tau. \end{cases} \tag{1}$$

The score function $S(\boldsymbol{x})$ is derived from an ID classification network $F$. $F$ can be further decomposed as a feature extractor $G$ sub-network and a linear layer ($\mathbf{W} \in R^{C \times D}, \boldsymbol{b} \in R^C$):

$$\boldsymbol{v} = F(\boldsymbol{x}) = \mathbf{W} \cdot G(\boldsymbol{x}) + \boldsymbol{b}, \qquad \boldsymbol{f} = G(\boldsymbol{x}), \tag{2}$$

where $\boldsymbol{f} \in R^D$ is the feature vector of the penultimate layer. Typically, network $F$ is trained with an ID training set using the standard cross-entropy loss $L_{CE}$:

$$L_{CE}(\boldsymbol{v}, \boldsymbol{y}) = \boldsymbol{y}^T \log(\sigma(\boldsymbol{v})), \qquad \boldsymbol{v} = F(\boldsymbol{x}), \tag{3}$$

where $\sigma$ is the softmax function and $\boldsymbol{v} \in R^C$ is the output logit.

Post-hoc OOD detection methods (Hendrycks et al., 2022; Liu et al., 2020; Djurisic et al., 2023) use pre-trained networks, off-the-shelf to feed directly into the scoring function. They focus on post-hoc adjustment to the features $\boldsymbol{f}$ and/or designing more effective score functions $S(\boldsymbol{x})$. On the other hand, training-based methods train a novel $F$ from scratch to improve the ID/OOD separation, *e.g.* by adding regularizers (Pinto et al., 2022; Wei et al., 2022) or data augmentations (Hendrycks et al., 2020b; Cubuk et al., 2020). They still require a compatible $S(\boldsymbol{x})$ to maximize the potential of $F$.

Typical scoring functions are based on the logits $\boldsymbol{v}$ or the features $\boldsymbol{f}$, or a combination of the two. For example, the maximal logit score (MLS) (Hendrycks et al., 2022) and energy-based score (Liu et al., 2020)are defined respectively as as:

$$S_{MLS}(\boldsymbol{x}) = \max_{j=1,\ldots,C} \boldsymbol{v}[j], \qquad S_{EBO}(\boldsymbol{x}) = log(\sum_{j=1}^{C} e^{\boldsymbol{v}[j]}) \tag{4}$$

where "$[j]$" denotes get the $j$-th element of the logit prediction. $S_{EBO}$ is a soft approximation of $S_{MLS}$, and other logit-based scores such as ASH (Djurisic et al., 2023) or FSEBO (Guan et al., 2024) are also related to $S_{MLS}$ since they modified logits by reshaping feature activation. A typical feature-based scoring function is the $k$-th nearest neighbor distance score (KNN) (Sun et al., 2022),

$$S_{KNN}(x) = -||\boldsymbol{f} - \boldsymbol{f}_{k^*}||_2, \tag{5}$$

where $f_{k^*}$ denotes the feature of the $k$-th nearest neighbor in the training set. The nearest neighbor guidance score (NNGuide) (Park et al., 2023) combines both features and logits, computed as:

$$S_{NNGuide}(\boldsymbol{x}) = S_{EBO}(\boldsymbol{x}) \cdot \text{Guide}(\boldsymbol{x}), \qquad \text{Guide}(\boldsymbol{x}) = \frac{1}{k} \sum_{i=1}^{k} S_{EBO}(\boldsymbol{x}_{i^*}) \cdot \cos(G(\boldsymbol{x}_{i^*}), \boldsymbol{f}), \tag{6}$$

where $x_{i^*}$ denotes the $i$-th nearest sample in the training set and $\cos(\cdot)$ the cosine similarity function.

## 3.2 Data-Augmentation Strategies

Depending on whether transform the label $y$, data augmentations can be categorized into data-based augmentation and label-based augmentation. In this work, we consider Random Erasing (RE) and Trivial Augment (TA) to represent data-based augmentation strategies, and label smoothing and mixup to represent label-based augmentation strategies. We select these four strategies because they are the additional strategies used by the `torchvision` v2 models (Torchvision, 2024) compared to `torchvision` v1 models.

**Data-Based Augmentation 1: Random Erasing (RE)** (Zhong et al., 2020) applies random zero masking in the input sample $x$ with a probability $p^{er}$. It reduces over-fitting and improve the generalization of neural networks. Typically, $p^{er} = 0.1$.

**Data-Based Augmentation 2: Trivial Augment (TA)** (Müller & Hutter, 2021) is a parameter-free set of image transformations to the input sample $x$ such as solarize, posterize, brightness adjustment, *etc*. During training, TA randomly selects a single augmentation and an augmentation strength from a pre-defined set.

**Label-Based Augmentation 1: Label Smoothing (LS)** (Szegedy et al., 2016) is used to avoid overconfidence by adding a uniform vector to the label $y$:

$$L_{CE}^{ls}(\boldsymbol{v}, \boldsymbol{y}^{ls}) = (\boldsymbol{y}^{ls})^T \log(\sigma(\boldsymbol{v})), \qquad \boldsymbol{y}^{ls} = (1 - \beta)\boldsymbol{y} + \beta\boldsymbol{u}, \qquad 0 \le \beta < 1, \tag{7}$$

where $\boldsymbol{u} \in R^C$ is a uniform vector with all elements equal to 1, $\beta$ is the label smoothing strength, and $\sigma$ is the softmax function. A larger $\beta$ denotes smoother learning targets; typically, $\beta = 0.1$.

**Label-Based Augmentation 2: Mixup** (Zhang, 2017) interpolates new samples $(x^{mix}, y^{mix})$ by linearly combining two samples in both the data and label spaces:

$$\boldsymbol{x}^{mix} = (1 - \lambda)\boldsymbol{x} + \lambda\boldsymbol{x}_1, \qquad \boldsymbol{y}^{mix} = (1 - \lambda)\boldsymbol{y} + \lambda\boldsymbol{y}_1. \tag{8}$$

The cross-entropy loss is applied to the mixed samples $(x^{mix}, y^{mix})$ in a standard fashion:

$$L_{CE}^{mix}(\boldsymbol{v}^{mix}, \boldsymbol{y}^{mix}) = (\boldsymbol{y}^{mix})^T \log(\sigma(\boldsymbol{v}^{mix})), \qquad \boldsymbol{v}^{mix} = F(\boldsymbol{x}^{mix}). \tag{9}$$

Mixup creates a smooth transition between different classes and can improve ID generalization.

## 4 Diagnosing Data Augmentations

In this section, we systematically investigate the influence of data augmentations on OOD detection. Starting with a case study in Sec. 4.1, we find that label-based data augmentations, label smoothing and mixup, harm OOD detection. Then, in Sec. 4.2, we explain the reason with a derivation. Specifically, label smoothing and mixup reduce the maximal logits of sample outputs, but more so for ID samples than OOD samples, leading to poorer ID/OOD separation. Furthermore, we analyze mixup from the perspective of virtual sample generation in Sec. 4.3. Empirical results suggest that adding mixup reduces the distinction between ID and mixed samples. Less separable ID and mixed samples will result in poor ID/OOD separation because mixed samples are close to OOD samples.

### 4.1 An Empirical Case Study Based on `torchvision`

The contributions in this paper are motivated by a case study based on the protocols of OpenOOD V1.5 (Lu et al., 2024). OpenOOD V1.5 is currently the largest OOD detection benchmark. The findings released by the authors are in line with previous literature showing the correlation between ID and OOD performance. A curious discrepancy that we noticed is that state-of-the-art methods for OOD almost all rely on `torchvision` v1 models. Yet the v1 is a basic model that lags in ID performance compared to v2 models with the same backbones. The v2 models improve ID performance by incorporating improved training techniques such as label smoothing and mixup.

We begin by comparing the performance of the v1 and v2 models using a ResNet50 backbone in ImageNet-1k (Deng et al., 2009). ResNet50-v2 improves accuracy by 4% compared to ResNet50-v1 but results in a 20% decrease in the OOD AUROC (see Fig. 1). Such a change in the OOD AUROC is significant because it surpasses the improvements that most post-hoc OOD methods achieve (Liu et al., 2020; Djurisic et al., 2023). Similar trends hold for other v1 and v2 models (see Fig. 1).

One of the key differences between v2 and v1 lies in the different data augmentation schemes. v2 uses both data- and label-based augmentations (see Sec. 3.2) on top of the simple augmentations (*e.g.* random resizing, cropping, and horizon flipping) used by v1. To pinpoint the influence of each augmentation strategy, we train models from scratch on ImageNet200 (Lu et al., 2024) with a single augmentation. Figure 2 compares the ID vs. OOD accuracy based on the MLS score, KNN score and NNGuide. More experimental details and results for CIFAR10/100 (Krizhevsky, 2009) are given in Appendix.

The impact of the augmentations is split. The data-only augmentations, *i.e.*, the Random Erasing (RE) and the Trivial Augment (TA), have minimal impact on the OOD accuracy, while the label-based augmentations, *i.e.*, label smoothing (LS) and mixup greatly decrease OOD performance. The effects likely compound together into the significant drop in OOD for v2 (all augs). These trends are most prominent at the logits, where the MLS scores (and NNGuide) are derived, but less pronounced at the feature level, where the KNN score is computed.

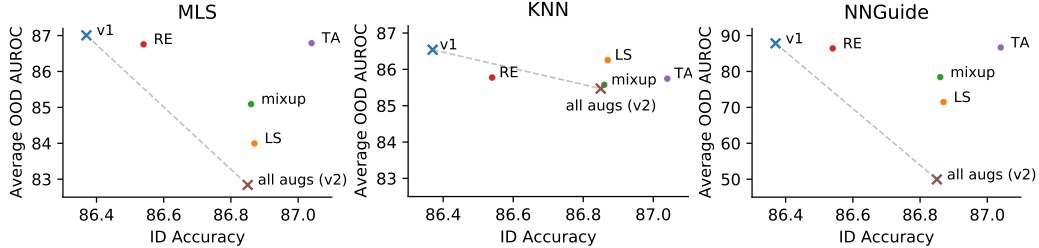

Figure 2: ID Accuracy and OOD detection AUROC for various data augmentations on ImageNet200, using a ResNet18 backbone as per (Yang et al., 2022).

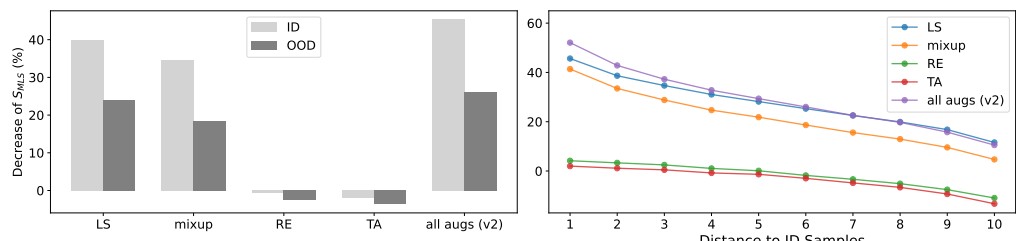

Figure 3: Relative decrease of the maximal logit $S_{MLS}$ with different data augmentations.

## 4.2 GRADIENT ANALYSIS OF DATA AUGMENTATIONS

This section analyzes how label smoothing and mixup influence OOD detection with the MLS scoring function $S_{MLS}$ (see equation 4). Proposition 4.1 shows that adding label smoothing and mixup will decrease the maximal logits $S_{MLS}$. Proposition 4.1 further shows that the decrement of $S_{MLS}$ is more pronounced for ID samples than OOD samples.

**Proposition 4.1.** *Let $i^*$ denote the index of the maximal logit, $\Delta\boldsymbol{v}[i^*]$ denote the increment of the maximal logit after one-step gradient descent, $L_{CE}$, $L_{CE}^{ls}$ and $L_{CE}^{mix}$ are defined as equation 3,7, and 9. We have*

$$\Delta\boldsymbol{v}[i^*] - \Delta\boldsymbol{v}^{ls/mix}[i^*] \propto \left(\frac{\partial L_{CE}^{ls/mix}}{\partial \boldsymbol{v}} - \frac{\partial L_{CE}}{\partial \boldsymbol{v}}\right)[i^*] \geq 0, \tag{10}$$

*where "$\sigma$" denotes the softmax function, and "$[j]$" denotes take the $j$-th element of a vector.*

**Remark:** Proposition 4.1 suggests that label smoothing and mixup tend to decrease the gradient updation to the maximal logits during each step, thus decreasing $S_{MLS}$. Detailed proof can be found in the appendix. Figure 3 (left) visualizes the decrement of $S_{MLS}$ of ID and OOD after data augmentations in ImageNet200. It can be observed that LS and Mixup will reduce $S_{MLS}$, while RE and TA do not significantly influence $S_{MLS}$.

To understand why LS and mixup reduce OOD performance, we delve into the $S_{MLS}$ decrement among ID and OOD samples in Proposition 4.2.

**Proposition 4.2.** *Let $i^*$ denote the index of the maximal logit, $\Delta v[i^*]$ denote the increment of the maximal logit after one-step gradient descent, $x^{id}/x^{ood}$ denote ID/OOD samples, $f^{id}/f^{ood}$ denote ID/OOD features, and $cos(\cdot, \cdot)$ denote the cosine similarity function. Assume the features $f$ are already learned while only the last fully-connected layer requires training; the feature norms of ID and OOD samples follow the same distribution, while the cosine similarity among features satisfies $\mathbb{E}\{cos(f_i^{id}, f_j^{id})\} \geq \mathbb{E}\{cos(f_i^{id}, f_j^{ood}))\}$. We have*

$$\mathbb{E}_{x^{id}}\{\Delta v[i^*] - \Delta v^{ls/mix}[i^*]\} \geq \mathbb{E}_{x^{ood}}\{\Delta v[i^*] - \Delta v^{ls/mix}[i^*]\}, \tag{11}$$

**Remark:** Proposition 4.2 shows that the $S_{MLS}$ *decreases more on the ID than OOD*, thus reducing the separability between ID and OOD samples. This result is experimentally verified and shown in Figure 3. Furthermore, we observed that *the decrement of $S_{MLS}$ is negatively correlated with the distance to the ID training set when adding label smoothing or mixup.* This also suggests that ID samples which should be closer to training samples in the feature space than OOD samples, will have larger decrements in $S_{MLS}$.

### 4.3 ANALYSIS OF MIXUP AS VIRTUAL SAMPLE GENERATION

Different from label smoothing, mixup creates virtual samples from ID data. We compare the maximal logit of mixed samples to that of ID and OOD samples in Figure 4. We find that: $i$) With the increasing $\lambda$, the AUROC becomes lower between mixed and OOD samples while higher between mixed and ID samples, meaning that the mixed samples will be inseparable from OOD samples. This suggests that the mixed samples can also serve as virtual OOD samples. $ii$) After adding mixup to the basic recipe v1, the AUROC of mixed and OOD samples will decrease for each $\lambda$, indicating that *adding mixup decreases the separability between ID and mixed samples.* As mixed samples get closer to OOD samples, less separable ID and mixed samples will likely cause less separability between ID and OOD.

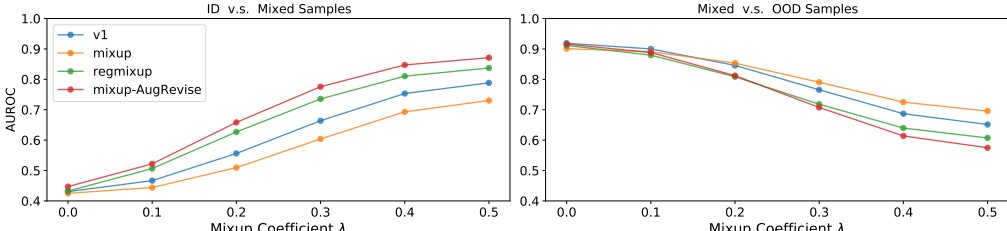

Figure 4: AUROC between ID/OOD and mixed samples with different mixup coefficient $\lambda$.

To sum up, label smoothing and mixup reduce the distinction between ID and OOD in logits during gradient updation. The negative influence will also be propagated into the feature space, as shown Figure 2. Compared to the impact on logits, the impact on the feature space is much smaller.

## 5 FIXING DATA AUGMENTATION FOR OUT-OF-DISTRIBUTION DETECTION

Based on the analysis of label smoothing and mixup, we devise two methods for fixing impaired logits. The first, augmentation deletion (Sec. 5.1), fixes the impaired logits by finetuning the last fully connected layer without problematic data augmentations. The second, augmentation revision (Sec. 5.2), revises the problematic data augmentations in the `torchvision` v2 receipt for training models from scratch.

### 5.1 AUGMENTATION DELETION (AUGDELETE) FOR PRETRAINED MODELS

Empirically, the impact of label smoothing and mixup is the greatest on the output logits. The effects gradually diminish with back-propagation into the feature layers. The results of figure 2 show less impact on OOD detection when adopting a feature-based score $S_{KNN}$ rather than a logit-based score $S_{MLS}$. These empirical results suggest that a simple way to fix the logits $v$ is to fine-tune the last fully connected layer $W, b$ without label smoothing and mixup.

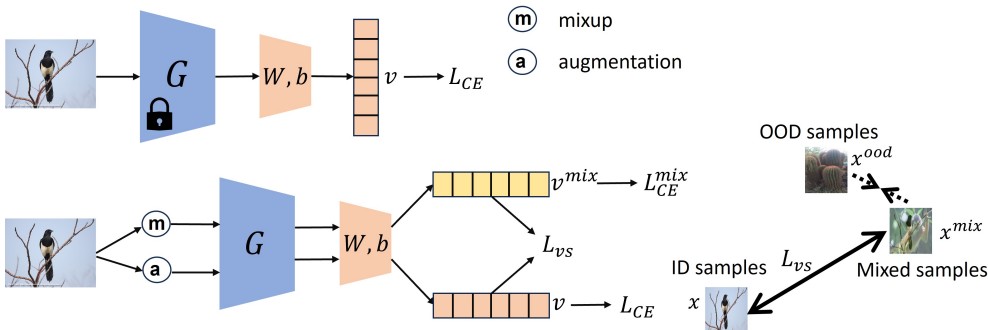

Figure 5: Pipelines of AugDelete (top) and AugRevise (bottom). In AugRevise, $L_{vs}$ is added to enforce the separation between ID and mixed samples. As mixed samples are close to OOD samples, better separation between ID and mixed samples can deliver better ID/OOD separation.

To make the finetuning process efficient, we extract the features $f$ in a single forward pass and then train $W, b$ with extracted $f$. Alg. 1 and Figure 5 show the pipeline of this simple approach termed as AugDelete. AugDelete can improve the logit-based OOD detectors with minimal training cost and maintain the ID accuracy since the feature extractor $G$ is fixed.

By retraining the last layer, AugDelete improves `torchvision` v2 models in terms of OOD detection. However, its OOD performance is simply comparable to v1 models (see ResNet or RegNet in Fig. 1) as the features themselves are left untouched. Next, we aim to surpass the v1 models in both ID and OOD by revising the v2 training recipe when training models from scratch.

## 5.2 AUGMENTATION REVISION (AUGREVISE) FOR MODELS TRAINED FROM SCRATCH

We follow the analysis from Section 4 and make the following design decisions. First, the data-based augmentations (Random Erasure and Trivial Augment) do not harm OOD detection, so they can be kept. Secondly, we remove label smoothing, since it harms OOD detection. Finally, we adjust the mixup scheme to ensure that ID samples are sufficiently separable from the mixed samples. Ideally, $S_{MLS}$ of ID samples should be larger than mixed samples. The closer the $\lambda$ to $0.5$, the greater the gap in $S_{MLS}$ between ID and mixed samples.

To improve mixup for OOD detection, Pinto et al. (2022) propose regmixup, which treats mixup loss as an OOD regularizer as

$$L_{CE}^{rmix}(\boldsymbol{v}^{mix}, \boldsymbol{y}^{mix}) = L_{CE}(\boldsymbol{v}, \boldsymbol{y}) + L_{CE}^{mix}(\boldsymbol{v}^{mix}, \boldsymbol{y}^{mix}). \tag{12}$$

However, we find that regmixup cannot ensure that ID samples are separable from that of mixed samples, as shown in Figure 4. As mixed samples are close to OOD samples, poor separation between ID and mixed samples will degrade ID/OOD separation. To ensure a clear separation between mixed and ID samples, we propose a virtual separation loss $L_{vs}$:

$$L_{vs}(\boldsymbol{v}, \boldsymbol{v}^{mix}, \lambda) = -(1 - P_\lambda) \log\left(\frac{\sum_{i=1}^{C} e^{\boldsymbol{v}[i]}}{\sum_{i=1}^{C} e^{\boldsymbol{v}[i]} + e^{\boldsymbol{v}^{mix}[i]}}\right) - P_\lambda \log\left(\frac{\sum_{i=1}^{C} e^{\boldsymbol{v}^{mix}[i]}}{\sum_{i=1}^{C} e^{\boldsymbol{v}[i]} + e^{\boldsymbol{v}^{mix}[i]}}\right), \tag{13}$$

$$P_\lambda = \frac{max(\lambda, 1 - \lambda)}{max(\lambda, 1 - \lambda) + 1}, \tag{14}$$

$L_{vs}$ optimize the LogSumExp(LSE) approximation of $S_{MLS}$ since this approximation provides dense gradients. It ensures the ratios between the maximal logits of ID and mixed samples ($\frac{S_{MLS}^{id}}{S_{MLS}^{mixup}}$) equals $\frac{1}{max(\lambda, 1-\lambda)}$. $\frac{S_{MLS}^{id}}{S_{MLS}^{mixup}} \geq 1$ ensures that $S_{MLS}$ of ID samples is larger than that of OOD samples. Moreover, $\frac{S_{MLS}^{id}}{S_{MLS}^{mixup}}$ increases as $\lambda$ becomes closer to $0.5$, ensuring the increasing distinction between mixed and ID samples. Overall, the final revised mixup adopts the loss $L_{CE}^{rvmix}$:

$$L_{CE}^{rvmix}(\boldsymbol{v}^{mix}, \boldsymbol{y}^{mix}, \lambda) = L_{CE}^{rmix}(\boldsymbol{v}^{mix}, \boldsymbol{y}^{mix}) + L_{vs}(\boldsymbol{v}, \boldsymbol{v}^{mix}, \lambda), \tag{15}$$

This augmentation revision approach is termed as AugRevise, the pipeline of which is shown in Alg. 2 and Figure 5. Note that AugRevise still requires AugDelete after training the whole network to mitigate the influence of data augmentation in the fully connected layers.

| **Algorithm 1: AugDelete** | **Algorithm 2: AugRevise** |
|---|---|
| **Input:** ID training set $\{\boldsymbol{x}_i, \boldsymbol{y}_i\}$, pre-trained network with $G, \mathbf{W}, \boldsymbol{b}$ | **Input:** ID training set $\{\boldsymbol{x}_i, \boldsymbol{y}_i\}$, initialized network with $G, \mathbf{W}, \boldsymbol{b}$ |
| **Output:** Finetuned linear layer $\mathbf{W}, \boldsymbol{b}$ | **Output:** Trained Network $G, \mathbf{W}, \boldsymbol{b}$ |
| 1: Extract features $\boldsymbol{f}_i$ with $G$ as equation 2 | 1: **while** Training not end **do** |
| 2: **while** Training not end **do** | 2:    Sample a batch of $(\boldsymbol{x}_i, \boldsymbol{y}_i)$ |
| 3:    Sample a batch of $(\boldsymbol{f}_i, \boldsymbol{y}_i)$ | 3:    Perform mixup to get $(\boldsymbol{x}_i^{mix}, \boldsymbol{y}_i^{mix})$ |
| 4:    Compute $L_{CE}$ as eq. equation 3 | 4:    Perform other data augmentations to $\boldsymbol{x}_i$ |
| 5:    Perform gradient descent to update $\mathbf{W}, \boldsymbol{b}$ | 5:    Compute logits $\boldsymbol{v}_i, \boldsymbol{v}_i^{mix}$ as equation 3, 8, 9 |
| 6: **end while** | 6:    Compute $L_{CE}^{rvmix}$ as equation 13~15 |
|  | 7:    Perform gradient descent to update $G, \mathbf{W}, \boldsymbol{b}$ |
|  | 8: **end while** |
|  | 9: Call AugDelete to update linear layer $\mathbf{W}, \boldsymbol{b}$ |
| 7: **return** $\mathbf{W}, \boldsymbol{b}$ | 10: **return** $G, \mathbf{W}, \boldsymbol{b}$ |

## 6 EXPERIMENTS

### 6.1 ABLATION STUDIES

We do ablation studies on ImageNet200 to verify critical elements of AugDelete and AugRevise on OOD Detection. By default, the maximal logit score $S_{MLS}$ is chosen as the OOD score function.

**AugDelete for Different Data Augmentation.** Figure 6 shows the OOD detection results before and after applying AugDelete under various data augmentations. We observe that AugDelete improves models with label smoothing and mixup by a large margin while maintaining the ID accuracy. AugDelete can also slightly improve the OOD Detection performance of RE and TA. However, with AugDelete, models trained with label smoothing and mixup are still worse than the v1 model. This is because AugDelete keeps the pretrained features, thus the negative impact of label smoothing and mixup are not mitigated.

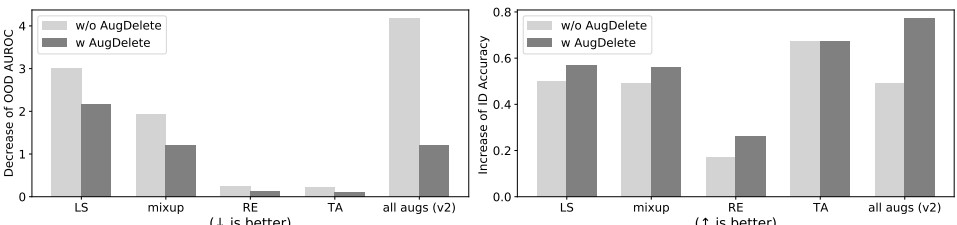

Figure 6: AugDelete for models trained with different data augmentations on ImageNet200

**Fixing Mixup for OOD Detection.** Mixup is fixed in AugRevise with $L_{vs}$ loss to increase the separability between ID and mixed samples. Table 1 shows the quantitative results of fixing mixup. Regmixup improves the vanilla mixup but cannot outperform the v1 model in OOD detection. Adopting mixup in AugRevise can outperform the v1 model in both ID classification and OOD detection. To explain the superior OOD performance of mixup-AugRevise to regmixup, we visualize the separability between ID, OOD, and mixed samples in Figure. 4. Mixup-AugRevise delivers higher auroc between ID and mixed samples, while lower auroc between mixed and OOD samples. It suggests better separation between ID and OOD samples and mixed samples as better virtual OOD samples, thus improving the separation between ID and OOD samples.

**Compare AugDelete and AugRevise** We compare AugDelete and AugRevise in Table 2. AugRevise outperforms AugDelete and vanilla v1 models in both the ID classification and OOD

Table 1: **Fixing mixup on ImageNet200.**

| Training Recipe | Loss | AUROC ↑ | FPR@95 ↓ | ID ACC ↑ |
|---|---|---|---|---|
| v1 | $L_{CE}$ | 87.00 | 46.90 | 86.37 |
| v1+mixup | $L_{CE}^{mix}$ | 84.00 | 57.81 | 86.87 |
| v1+regmixup | $L_{CE}^{rmix}$ | 86.97 | 48.03 | 87.58 |
| v1+mixup-AugRevise | $L_{CE}^{rvmix}$ | 87.72 | 42.09 | 87.28 |

Table 2: **Comparison on ImageNet200.**

| Training Recipe | AUROC ↑ | FPR@95 ↓ | ID ACC ↑ |
|---|---|---|---|
| v1 | 87.00 | 46.90 | 86.37 |
| v2 | 82.84 | 63.05 | 86.89 |
| v2+AugDelete | 85.81 | 51.74 | 87.14 |
| AugRevise | 87.88 | 41.72 | 87.67 |
| AugRevise+LS | 87.17 | 43.87 | 87.33 |

detection. However, adding label smoothing in AugRevise will decrease the OOD performance of OOD detection, suggesting that label smoothing should be removed in AugRevise.

## 6.2 AUGDELETE FOR PRETRAINED MODELS OF IMAGENET-1K IN TORCHVISION

**OOD detection with Various Pretrained Network Architectures.** We apply AugDelete to pre-trained models with different network architectures including convolutional neural networks (CNNs) and transformers. Figure 1 visualizes the ID accuracy and OOD performance with/without AugDelete. We see that AugDelete improves the OOD detection of both CNNs and transformers while maintaining ID accuracy. Besides, models with better ID accuracy show higher or at least comparable AUROC after applying AugDelete.

**AugDelete for various OOD score functions.** We apply AugDelete with various OOD score functions $S(\boldsymbol{x})$ using torchvision v2 pretrained models. Both logit-based (MLS,EBO,ASH), feature-based (KNN) and combing both scores (NNGuide) are considered. Table 3 shows the results with torchvision models, including ResNet50-v1 and ResNet50-v2 and ResNet50-v2+AugDelete. We can see that AugDelete can improve ResNet50-v2 in all $S(\boldsymbol{x})$ by a large margin, except KNN scores since features are not changed in AugDel. AugDelete performs comparably to ResNet50-v1 with logit-based $S(\boldsymbol{x})$ in terms of OOD detection, while having much better ID accuracy than ResNet50-v1. However, AugDelete shows worse OOD detection performance than ResNet50-v1 when adopting feature-based $S(\boldsymbol{x})$, KNN or NNGuide. This is because AugDel does not fix the impaired features of ResNet50-v2.

Table 3: **AugDelete for various OOD score functions on ImageNet-1k.**

| Method | ResNet50-v1 | | | ResNet50-v2 | | | ResNet50-v2 + AugDelete | | |
|---|---|---|---|---|---|---|---|---|---|
| | AUROC ↑ | FPR@95 ↓ | ID ACC ↑ | AUROC ↑ | FPR@95 ↓ | ID ACC ↑ | AUROC ↑ | FPR@95 ↓ | ID ACC ↑ |
| MLS (Hendrycks et al., 2022) | 83.02 | 53.02 | 76.18 | 72.84 | 84.75 | 80.92 | 83.08 | 63.11 | 80.31 |
| EBO (Liu et al., 2020) | 82.68 | 53.48 | 76.18 | 52.88 | 89.97 | 80.92 | 81.83 | 65.59 | 80.31 |
| ASH (Djurisic et al., 2023) | 83.97 | 49.62 | 76.18 | 53.53 | 90.59 | 80.92 | 81.70 | 65.11 | 80.31 |
| KNN (Sun et al., 2022) | 80.64 | 52.50 | 76.18 | 79.91 | 55.09 | 80.92 | 79.91 | 55.09 | 80.31 |
| NNGuide (Park et al., 2023) | 86.68 | 44.81 | 76.18 | 65.77 | 72.07 | 80.92 | 77.54 | 58.22 | 80.31 |

## 6.3 AUGREVISE FOR TRAINING-TIME MODEL ENHANCEMENT

We train models from scratch with AugRevise on ImageNet200/1k and CIFAR10/100 datasets. Following the same training setting as OpenoodV1.5, all the AugRevise models are trained for 100 epochs with learning rate starts from 0.1. ResNet18 is adopted for CIFAR10/100 and ImageNet200, while ResNet50 is for ImageNet200. We choose logit-based (MLS), feature-based (KNN), and logit and a combination of both (NNGuide) OOD score functions for AugRevise. Table 4 and 5 compares AugRevise with state-of-the-art (SOTA) methods in Openood V1.5 Benchmark. AugRevise improves both logit-based and feature-based methods since it improves both features and logits. AugRevise also improves ID accuracy and outperforms comparing methods. Overall, AugRevise outperforms both post-hoc and training-based methods in ID and OOD.

## 7 CONCLUSION

In this paper, we identify that certain widely used data augmentations, label smoothing and mixup, harm OOD detection despite improving ID classification. Through theoretical and empirical analysis, we find that label smoothing and mixup reduce the separation between OOD and ID data in the

Table 4: **Comparison with SOTA methods on CIFAR10/100.** Method are grouped as post-hoc and training-based OOD detection methods, respectively.

| Method | CIFAR10 | | | CIFAR100 | | |
|---|---|---|---|---|---|---|
| | AUROC ↑ | FPR@95 ↓ | ID ACC ↑ | AUROC ↑ | FPR@95 ↓ | ID ACC ↑ |
| v1+MLS (Hendrycks et al., 2022) | 89.31 | 51.50 | 95.06 | 80.36 | 56.09 | 77.26 |
| v1+EBO (Liu et al., 2020) | 89.39 | 51.51 | 95.06 | 80.34 | 56.09 | 77.26 |
| v1+MSP (Hendrycks & Gimpel, 2017b) | 89.38 | 39.95 | 95.06 | 79.02 | 56.75 | 77.26 |
| v1+ASH (Djurisic et al., 2023) | 89.34 | 51.42 | 95.06 | 80.50 | 55.84 | 77.26 |
| v1+FSEBO (Guan et al., 2024) | 88.08 | 59.00 | 95.06 | 79.97 | 57.24 | 77.26 |
| v1+KNN (Sun et al., 2022) | 91.80 | 29.14 | 95.06 | 81.29 | 57.44 | 77.26 |
| v1+NNGuide (Park et al., 2023) | 85.25 | 72.22 | 95.06 | 80.84 | 57.51 | 77.26 |
| T2FNorm+T2FNorm (Regmi et al., 2023) | 94.89 | 19.61 | 94.69 | 81.28 | 54.86 | 76.43 |
| LogitNorm+MSP (Wei et al., 2022) | 94.53 | 21.57 | 94.30 | 80.00 | 58.25 | 76.34 |
| VOS+EBO (Du et al., 2022) | 89.27 | 48.73 | 94.31 | 81.12 | 54.63 | 77.20 |
| NPOS+KNN (Tao et al., 2023) | 91.92 | 26.62 | — | 80.32 | 57.24 | — |
| CIDER+KNN (Ming et al., 2023) | 92.71 | 26.41 | — | 76.79 | 63.12 | — |
| MOS+MOS (Huang & Li, 2021) | 73.93 | 70.81 | 94.83 | 80.29 | 56.67 | 76.98 |
| AugMix+MSP (Hendrycks et al., 2020b) | 90.55 | 32.34 | 95.01 | 78.27 | 57.33 | 76.45 |
| RegMixup+MSP (Pinto et al., 2022) | 88.86 | 42.54 | 95.75 | 79.94 | 56.81 | 79.32 |
| AugRevise +MLS (Ours) | 94.03 | 25.27 | 96.73 | 83.69 | 50.86 | 82.10 |
| AugRevise +MSP (Ours) | 93.50 | 24.11 | 96.73 | 82.50 | 52.12 | 82.10 |
| AugRevise +KNN (Ours) | 94.95 | 21.83 | 96.73 | 83.88 | 53.46 | 82.10 |
| AugRevise +NNGuide (Ours) | 94.22 | 26.45 | 96.73 | 84.51 | 49.52 | 82.10 |

Table 5: **Comparison with SOTA methods on ImageNet200/1k.**

| Method | ImageNet200 | | | ImageNet-1k | | |
|---|---|---|---|---|---|---|
| | AUROC ↑ | FPR@95 ↓ | ID ACC ↑ | AUROC ↑ | FPR@95 ↓ | ID ACC ↑ |
| v1+MLS (Hendrycks et al., 2022) | 87.00 | 46.90 | 86.37 | 83.02 | 53.02 | 76.18 |
| v1+EBO (Liu et al., 2020) | 86.68 | 47.54 | 86.37 | 82.68 | 53.48 | 76.18 |
| v1+MSP (Hendrycks & Gimpel, 2017b) | 86.73 | 45.13 | 86.37 | 80.63 | 58.57 | 76.18 |
| v1+ASH (Djurisic et al., 2023) | 87.19 | 46.25 | 86.37 | 83.97 | 49.62 | 76.18 |
| v1+FSEBO(Guan et al., 2024) | 86.75 | 48.87 | 86.37 | 86.82 | 45.14 | 76.18 |
| v1+KNN (Sun et al., 2022) | 86.54 | 44.70 | 86.37 | 80.64 | 52.50 | 76.18 |
| v1+NNGuide | 87.83 | 46.90 | 86.37 | 86.68 | 44.81 | 76.18 |
| T2FNorm+T2FNorm (Regmi et al., 2023) | 88.28 | 40.37 | 86.87 | 82.50 | 50.19 | 76.76 |
| LogitNorm+MSP (Wei et al., 2022) | 87.85 | 40.28 | 86.04 | 83.08 | 49.94 | 76.45 |
| VOS+EBO (Du et al., 2022) | 86.75 | 46.95 | 86.23 | — | — | — |
| NPOS+KNN (Tao et al., 2023) | 86.94 | 41.93 | — | — | — | — |
| CIDER+KNN (Ming et al., 2023) | 85.62 | 45.14 | — | 80.58 | 50.19 | — |
| MOS+MOS (Huang & Li, 2021) | 75.15 | 61.58 | 85.60 | 77.80 | 64.47 | 72.81 |
| AugMix+MSP (Hendrycks et al., 2020b) | 87.09 | 44.20 | 87.01 | 82.08 | 55.70 | 77.63 |
| RegMixup+MSP (Pinto et al., 2022) | 87.47 | 49.62 | 87.25 | 81.68 | 57.12 | 76.68 |
| AugRevise +MLS (Ours) | 87.88 | 41.72 | 87.67 | 84.77 | 49.06 | 77.70 |
| AugRevise +MSP (Ours) | 87.89 | 41.65 | 87.67 | 84.78 | 49.06 | 77.70 |
| AugRevise +KNN (Ours) | 87.00 | 41.66 | 87.67 | 82.56 | 49.25 | 77.70 |
| AugRevise +NNGuide (Ours) | 89.31 | 37.56 | 87.67 | 87.17 | 43.64 | 77.70 |

logit space, thus hurting OOD detection. To mitigate the negative impact, we proposed AugDelete for finetuning pretrained models and AugRevise for training from scratch. Both approaches can improving OOD detection performance while maintaining strong ID accuracy.

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

## A    DERIVATION OF PROPOSITION 4.1

We use $i^*$ denote the index of the maximal logit, $\Delta \boldsymbol{v}[i^*]$ to denote increment of the maximal logit after one-step gradient descent, $L_{CE}$, $L_{CE}^{ls}$ and $L_{CE}^{mix}$ are defined as equation 3,7, and 9.

The derivation contains 2 steps. First, we illustrate the relationship between the one-step update of the maximal logit ($\Delta \boldsymbol{v}[i^*]$) and the gradient. Then, we compute the difference between gradients. With the results of the previous steps, we finally prove the proposition.

### A.1    RELATING THE INCREMENT OF THE MAXIMAL LOGIT TO GRADIENTS

We follow the loss and network definition as equation 3 and 2. Let $\theta$ denote the parameter of the feature extraction network $G$, and $\eta$ denote the learning rate. When applying one-step gradient descent, the network parameters $\mathbf{W}$, $\boldsymbol{b}$, and $\theta$ are directly updated, then the update of the network parameters will be reflected on the logits. According to the chain rule, the total derivative $\Delta \boldsymbol{v}[i^*]$ of the maximal logits $\boldsymbol{v}[i^*]$ is:

$$
\begin{aligned}
\Delta \boldsymbol{v}[i^*] &= \boldsymbol{f}^T \Delta \mathbf{W}[i^*,:] + \mathbf{W}[i^*,:]^T \Delta \boldsymbol{f} + \Delta \boldsymbol{b}[i^*] \\
&= \boldsymbol{f}^T \Delta \mathbf{W}[i^*,:] + \mathbf{W}[i^*,:]^T (\frac{\partial \boldsymbol{f}}{\partial \theta})^T \Delta \theta + \Delta \boldsymbol{b}[i^*] \\
&= \boldsymbol{f}^T (-\eta \frac{\partial L_{ce}}{\partial \mathbf{W}[i^*,:]}) + \mathbf{W}[i^*,:]^T (\frac{\partial \boldsymbol{f}}{\partial \theta})^T (-\eta \frac{\partial L_{ce}}{\partial \theta}) + (-\eta \frac{\partial L_{ce}}{\partial \boldsymbol{b}[i^*]}) \\
&= -\eta \{ \boldsymbol{f}^T \frac{\partial \boldsymbol{v}[i^*]}{\partial \mathbf{W}[i^*,:]} \frac{\partial L_{ce}}{\partial \boldsymbol{v}[i^*]} + \mathbf{W}[i^*,:]^T (\frac{\partial \boldsymbol{f}}{\partial \theta})^T (\frac{\partial \boldsymbol{f}}{\partial \theta} \sum_{k=1}^{C} \frac{\partial \boldsymbol{v}[k]}{\partial \boldsymbol{f}} \frac{\partial L_{ce}}{\partial \boldsymbol{v}[k]}) + \frac{\partial \boldsymbol{v}[i^*]}{\partial \boldsymbol{b}[i^*]} \frac{\partial L_{ce}}{\partial \boldsymbol{v}[i^*]} \} \\
&= -\eta \{ (\boldsymbol{f}^T \boldsymbol{f} + 1) \frac{\partial L_{ce}}{\partial \boldsymbol{v}[i^*]} + \sum_{k=1}^{C} \mathbf{W}[i^*,:]^T (\frac{\partial \boldsymbol{f}}{\partial \theta})^T (\frac{\partial \boldsymbol{f}}{\partial \theta}) \mathbf{W}[k,:] \frac{\partial L_{ce}}{\partial \boldsymbol{v}[k]}) \} \\
&\approx -\eta \{ \boldsymbol{f}^T \boldsymbol{f} + 1 + \mathbf{W}[i^*,:]^T (\frac{\partial \boldsymbol{f}}{\partial \theta})^T (\frac{\partial \boldsymbol{f}}{\partial \theta}) \mathbf{W}[i^*,:] \} \frac{\partial L_{ce}}{\partial \boldsymbol{v}[i^*]} \\
&\propto -\frac{\partial L_{ce}}{\partial \boldsymbol{v}[i^*]}
\end{aligned}
\tag{16}
$$

where "$[j]$" denotes take the $j$-th element of a vector, and "$[k,:]$" denotes take the $k$-th row of a matrix.

**Remark:** During training, the gradient of the maximal logits tends to be much larger than that of the other logits, *i.e.* $|\frac{\partial L_{ce}}{\partial \boldsymbol{v}[i^*]}| >> |\frac{\partial L_{ce}}{\partial \boldsymbol{v}[k]}|(k \neq i^*)$. Besides, $\mathbf{W}[i^*,:]^T \mathbf{W}[i^*,:]$ tends to be larger than $\mathbf{W}[i^*,:]^T \mathbf{W}[k,:](k \neq i^*)$. Based on these reasons, the approximation is reasonable.

According to equation 16, we have

$$
\begin{aligned}
\Delta \boldsymbol{v}[i^*] - \Delta \boldsymbol{v}^{ls/mix}[i^*] &\approx \eta (\boldsymbol{f}^T \boldsymbol{f} + 1)(\frac{\partial L_{ce}^{ls/mix}}{\partial \boldsymbol{v}[i^*]} - \frac{\partial L_{ce}}{\partial \boldsymbol{v}[i^*]}) \\
&\propto \frac{\partial L_{ce}^{ls/mix}}{\partial \boldsymbol{v}[i^*]} - \frac{\partial L_{ce}}{\partial \boldsymbol{v}[i^*]}
\end{aligned}
\tag{17}
$$

### A.2    DIFFERENCE OF GRADIANTS

For cross entropy loss $L_{ce}$ defined in equation 3, we can compute the partial derivative w.r.t the $j$-th logits $v[j]$ as:

$$
\frac{\partial L_{ce}}{\partial \boldsymbol{v}}[j] = -\boldsymbol{y}[j] + \boldsymbol{p}[j], \qquad \boldsymbol{p}[i] = \frac{e^{\boldsymbol{v}[j]}}{\sum_{k=1}^{C} e^{\boldsymbol{v}[k]}},
\tag{18}
$$

Similarly, the gradient for label smoothing w.r.t the $j$-th logits $v[j]$ is

$$
\frac{\partial L_{ce}^{ls}}{\partial \boldsymbol{v}}[j] = -\boldsymbol{y}^{ls}[j] + \boldsymbol{p}[j],
\tag{19}
$$

compare equation 18 and 19, we can get the gradient difference

$$\frac{\partial L_{ce}^{ls}}{\partial \boldsymbol{v}}[j] - \frac{\partial L_{ce}}{\partial \boldsymbol{v}}[j] = \boldsymbol{y}[j] - \boldsymbol{y}^{ls}[j], \tag{20}$$

For mixup, we adopt the first-order approximation derived by Zou et al. (2023), i.e.,

$$L_{ce}^{mix} \approx L_{ce} + (\boldsymbol{y} - \sigma(\boldsymbol{v}))^T \boldsymbol{v} \tag{21}$$

With equation 21, we can compute the gradient difference $\frac{\partial L_{ce}^{mix}}{\partial \boldsymbol{v}[j]} - \frac{\partial L_{ce}}{\partial \boldsymbol{v}[j]}$:

$$\frac{\partial L_{ce}^{mix}}{\partial \boldsymbol{v}}[j] - \frac{\partial L_{ce}}{\partial \boldsymbol{v}}[j] = (\boldsymbol{y}[j] - \boldsymbol{p}[j]) + p[j] \sum_{k=1}^{C} p[k](\boldsymbol{v}[k] - \boldsymbol{v}[j]) \tag{22}$$

Now we analyze the gradient of the maximal logit $\boldsymbol{v}[i^*]$. Take $j = i^*$ into equation 20 and combining the definition of label smoothing in equation 7, we have

$$\frac{\partial L_{ce}^{ls}}{\partial \boldsymbol{v}}[i^*] - \frac{\partial L_{ce}}{\partial \boldsymbol{v}}[i^*] = \beta(1 - \frac{1}{C}) > 0, \tag{23}$$

where $\beta$ is the label smoothing coefficient and $C$ is the number of classes. Similarly, take $j = i^*$ into equation 22, we have

$$\begin{aligned}
\frac{\partial L_{ce}^{mix}}{\partial \boldsymbol{v}}[i^*] - \frac{\partial L_{ce}}{\partial \boldsymbol{v}}[i^*] &= (1 - \boldsymbol{p}[i^*]) + p[i^*] \sum_{k=1}^{C} p[k](\boldsymbol{v}[k] - \boldsymbol{v}[i^*]) \\
&= \sum_{k=1,k\neq i^*}^{C} \boldsymbol{p}[k] + \sum_{k=1,k\neq i^*}^{C} p[i^*]p[k](\boldsymbol{v}[k] - \boldsymbol{v}[i^*]) \\
&= \sum_{k=1,k\neq i^*}^{C} \boldsymbol{p}[k] + p[i^*]p[k](\boldsymbol{v}[k] - \boldsymbol{v}[i^*]) \\
&= \sum_{k=1,k\neq i^*}^{C} \boldsymbol{p}[k] + p[i^*]p[k] \log(\frac{\boldsymbol{p}[k]}{\boldsymbol{p}[i^*]}) \\
&= \sum_{k=1,k\neq i^*}^{C} \boldsymbol{p}[k]\{1 + p[i^*] \log(\frac{\boldsymbol{p}[k]}{\boldsymbol{p}[i^*]})\} \geq 0
\end{aligned} \tag{24}$$

**Remark:** The informative gradients come from the wrong predicted logits, i.e., $p[k] \geq \boldsymbol{p}[i^*]$. When $p[k] \geq \boldsymbol{p}[i^*]$ holds, $\boldsymbol{p}[k]\{1 + p[i^*] \log(\frac{\boldsymbol{p}[k]}{\boldsymbol{p}[i^*]})\}$ is large than 0. On the contrary, for the correct predicted logits, $p[k] << \boldsymbol{p}[i^*]$, then the term $\boldsymbol{p}[k]\{1 + p[i^*] \log(\frac{\boldsymbol{p}[k]}{\boldsymbol{p}[i^*]})\}$ is close to 0.

Combining equation 17, 23, and 24, we can reach the conclusion:

$$\Delta \boldsymbol{v}[i^*] - \Delta \boldsymbol{v}^{ls/mix}[i^*] \propto \frac{\partial L_{ce}^{ls/mix}}{\partial \boldsymbol{v}[i^*]} - \frac{\partial L_{ce}}{\partial \boldsymbol{v}[i^*]} \geq 0. \tag{25}$$

## B  DERIVATION OF PROPOSITION 4.2

Let $\boldsymbol{x}$, $\boldsymbol{x}^{id}$ and $\boldsymbol{x}^{ood}$ denote ID train samples, ID test samples, and OOD test samples, and $\boldsymbol{f}$, $\boldsymbol{f}^{id}$ and $\boldsymbol{f}^{ood}$ be their corresponding features, respectively. $i^*$ is the index of the maximal logits. According to equation 16, we have:

$$\begin{aligned}
&\Delta \boldsymbol{v}_{id}[i^*] - \Delta \boldsymbol{v}_{id}^{ls/mix}[i^*] \\
&\approx \eta\{(\boldsymbol{f}^T \boldsymbol{f}^{id} + 1)\}(\frac{\partial L_{ce}^{ls/mix}}{\partial \boldsymbol{v}[i^*]} - \frac{\partial L_{ce}}{\partial \boldsymbol{v}[i^*]}) \\
&= \eta(||\boldsymbol{f}|| \cdot ||\boldsymbol{f}^{id}||cos(\boldsymbol{f}, \boldsymbol{f}^{id}) + 1)(\frac{\partial L_{ce}^{ls/mix}}{\partial \boldsymbol{v}[i^*]} - \frac{\partial L_{ce}}{\partial \boldsymbol{v}[i^*]})
\end{aligned} \tag{26}$$

$$\Delta \boldsymbol{v}_{ood}[i^*] - \Delta \boldsymbol{v}_{ood}^{ls/mix}[i^*]$$

$$\approx \eta\{(\boldsymbol{f}^T \boldsymbol{f}^{ood} + 1)\}(\frac{\partial L_{ce}^{ls/mix}}{\partial \boldsymbol{v}[i^*]} - \frac{\partial L_{ce}}{\partial \boldsymbol{v}[i^*]}) \quad (27)$$

$$= \eta(||\boldsymbol{f}|| \cdot ||\boldsymbol{f}^{ood}||cos(\boldsymbol{f}, \boldsymbol{f}^{ood}) + 1)(\frac{\partial L_{ce}^{ls/mix}}{\partial \boldsymbol{v}[i^*]} - \frac{\partial L_{ce}}{\partial \boldsymbol{v}[i^*]})$$

Since the norm of $\boldsymbol{f}^{id}$ and $\boldsymbol{f}^{ood}$ follow the same distribution, while $\mathbb{E}\{\mathbf{cos}(f_i^{id}, f_j^{id})\} \geq \mathbb{E}\{\mathbf{cos}(f_i^{id}, f_j^{ood}))\}$, we have

$$\mathbb{E}_{\boldsymbol{x}^{id}}\{||\boldsymbol{f}|| \cdot ||\boldsymbol{f}^{id}||cos(\boldsymbol{f}, \boldsymbol{f}^{id}))\} = ||\boldsymbol{f}|| \cdot \mathbb{E}_{\boldsymbol{x}^{id}}\{||\boldsymbol{f}^{id}||cos(\boldsymbol{f}, \boldsymbol{f}^{id}))\}$$
$$\geq \mathbb{E}_{\boldsymbol{x}^{ood}}\{||\boldsymbol{f}|| \cdot ||\boldsymbol{f}^{ood}||cos(\boldsymbol{f}, \boldsymbol{f}^{ood}))\} \quad (28)$$

Combing equation 26∼ 28, we reach

$$\mathbb{E}_{\boldsymbol{x}^{id}}\{\Delta \boldsymbol{v}[i^*] - \Delta \boldsymbol{v}^{ls/mix}[i^*]\} \geq \mathbb{E}_{\boldsymbol{x}^{ood}}\{\Delta \boldsymbol{v}[i^*] - \Delta \boldsymbol{v}^{ls/mix}[i^*]\}. \quad (29)$$

**Remark**: Since features $f$ is already trained, there is no gradient updation to $f$. As a result, equation 16 can simplified as $\Delta \boldsymbol{v}[i^*] \approx -\eta\{\boldsymbol{f}^T\boldsymbol{f} + 1\}\frac{\partial L_{ce}}{\partial \boldsymbol{v}[i^*]}$. Based on this simplification, we can derive equation 26 and 27.

## C   EXPERIMENT DETAILS

### C.1   OPENOODV1.5 DATASET SETTING

We conduct experiments on OpenOOD v1.5 (Zhang et al., 2023) benchmark. It consists of 4 ID datasets, CIFAR10/100 (Krizhevsky, 2009), ImageNet200 (Zhang et al., 2023), and ImageNet-1k (Deng et al., 2009). Each ID dataset contains several near-OOD and far-OOD test sets, where the near-OOD test sets are more challenging than the far-OOD ones.

**CIFAR10/100** are relatively small datasets. They have the same far-OOD test set containing MNIST (LeCun et al., 1998), SVHN (Netzer et al., 2011), Textures (Cimpoi et al., 2014), and Places365 (Zhou et al., 2018). TinyImageNet (Le & Yang, 2015) and CIFAR100 are adopted as near-OOD evaluation sets for CIFAR10; while TinyImageNet and CIFAR10 are adopted for CIFAR100. For network architecture, we adopt the same network backbone, ResNet18 as OpenoodV1.5 (Zhang et al., 2023).

**ImageNet-1k** is a large-scale dataset consisting of 1281167 training images of 1000 classes. It has 2 near-OOD dataset, SSB-hard (Vaze et al., 2022a) and NINCO (Bitterwolf et al., 2023), and 3 far-OOD datasets, iNaturalist (Horn et al., 2018), Textures (Cimpoi et al., 2014), and OpenImage-O (Wang et al., 2022). To be consistent with Openood V1.5 benchmark, ResNet50, Swin Transformer (Swin-T), and Vision Transformers (ViT) are adopted as pretrained backbone. Besides these models, some additional `torchvision` pre-trained models are evaluated.

**ImageNet200** is a subset of ImageNet-1k, which contains 200 classes. It has the same near-OOD and far-OOD datasets as ImageNet-1k. Following OpenoodV1.5 (Zhang et al., 2023), ResNet-18 is adopted as the network backbone.

**Evaluation Metrics.** FPR@95 and AUROC are adopted to evaluate the OOD performance. FPR@95 is the false positive rate when the true positive rate is 95%, while AUROC is the Area under the receiver operating characteristic curve. Lower FPR@95 and higher AUROC deliver better separation between ID and OOD samples.

### C.2   TRAINING SETTINGS

We follow the training setting of Openoodv1.5 (Zhang et al., 2023). All models are trained 100 epochs with learning rates starting from 0.1. The same Cosine learning rate schedule is adopted as OpenOODv1.5. For CIFAR10/100 and ImageNet200, batch size is 128; for ImageNet-1k, batch size is 512. All the models are repeated with 3 random seeds and the mean results are reported.

For AugDelete models, we retrain the fc layers for 15 epochs with learning rates starting from 0.01. Following `torchvision`, both vanilla mixup (Zhang, 2017) and cutmix (Yun et al., 2019) are adopted for models with mixup. The mixup strength $\lambda$ of mixup and cutmix follows Beta distribution $Beta(0.2, 0.2)$ and $Beta(1, 1)$, respectively.

We compare our baseline results (receipt v1) with the OpenOOD v1.5 checkpoint with receipt v1 in Table 6. All the models are repeated with 3 random seeds and the mean results are reported. Our re-implementation shows similar results as OpenOOD v1.5.

Table 6: **Comparison between OpenoodV1.5 models and our re-implementation.** The same receipt v1 is adopted in both implementations.

| Dataset | Implementation | Near-OOD | | Far-OOD | | ID ACC |
| | | AUROC ↑ | FPR@95 ↓ | AUROC ↑ | FPR@95 ↓ | ↑ |
|---|---|---|---|---|---|---|
| CIFAR10 | Ours | 87.54 | 60.39 | 91.04 | 40.17 | 94.59 |
| | Openood V1.5 | 87.52 | 61.32 | 91.10 | 41.68 | 95.06 |
| CIFAR100 | Ours | 81.16 | 55.69 | 80.75 | 54.49 | 77.22 |
| | Openood V1.5 | 81.05 | 55.47 | 79.67 | 56.73 | 77.25 |
| ImageNet200 | Ours | 82.43 | 60.21 | 90.84 | 34.40 | 86.40 |
| | Openood V1.5 | 82.90 | 59.76 | 91.11 | 34.03 | 86.37 |

## C.3 Imapact of Data Augmentation on OOD Detection

Table 7, 8 and 9 show the influence of each data augmentation on CIFAR10/100 and ImageNet200, with a single augmentation for each time. Follow the same training setting as (Yang et al., 2022), each training configure is repeated 3 times, and mean results are reported. Besides the MLS score function, we also adopt the KNN score (Sun et al., 2022) and NNGuide (Park et al., 2023) to reflect the OOD detection ability of the feature space.

Table 7: **OOD detection results w.r.t data augmenations on CIFAR10.**

| Score | Data Augmentaion | Near-OOD | | Far-OOD | | ID ACC |
| | | AUROC ↑ | FPR@95 ↓ | AUROC ↑ | FPR@95 ↓ | ↑ |
|---|---|---|---|---|---|---|
| MLS | v1 | 87.52 | 61.32 | 91.10 | 41.68 | 95.06 |
| | v1 + LS | 82.13 | 93.76 | 86.05 | 85.40 | 95.23 |
| | v1 + mixup | 84.39 | 76.20 | 88.89 | 58.10 | 95.95 |
| | v1 + RE | 88.47 | 56.22 | 91.99 | 39.53 | 95.36 |
| | v1 + TA | 92.15 | 32.37 | 95.28 | 19.68 | 95.51 |
| | v1+all augs (v2) | 85.88 | 77.50 | 92.07 | 45.65 | 95.86 |
| KNN | v1 | 90.64 | 33.99 | 92.96 | 24.28 | 95.06 |
| | v1+LS | 90.03 | 36.81 | 93.12 | 21.48 | 95.23 |
| | v1+mixup | 91.58 | 33.38 | 94.33 | 21.47 | 95.95 |
| | v1+RE | 91.30 | 33.01 | 93.96 | 22.15 | 95.36 |
| | v1+TA | 92.32 | 28.88 | 95.26 | 18.83 | 95.51 |
| | v1+all augs (v2) | 92.82 | 27.92 | 96.34 | 16.84 | 95.86 |
| NNGuide | v1 | 83.54 | 78.57 | 86.95 | 65.86 | 95.06 |
| | v1+LS | 78.34 | 85.27 | 81.70 | 74.45 | 95.23 |
| | v1+mixup | 78.21 | 80.22 | 83.34 | 66.39 | 95.95 |
| | v1+RE | 84.42 | 73.46 | 88.83 | 58.68 | 95.36 |
| | v1+TA | 90.32 | 44.11 | 94.46 | 26.41 | 95.51 |
| | v1+all augs (v2) | 81.48 | 72.87 | 92.60 | 35.96 | 95.86 |

## C.4 AugDelete for Different Data Augmentation

Table 10, 11 and 12 shows the OOD detection results before and after applying AugDelete under various data augmentations. On all 3 datasets, We observe that AugDelete improves models with label smoothing and mixup by a large margin while maintaining the ID accuracy. AugDelete can also slightly improve the OOD Detection performance of RE and TA. However, with AugDelete, models trained with mixup or LS are still worse than the v1 model. This is because AugDelete keeps the pretrained features, thus the negative impact of label smoothing and mixup are not mitigated.

Table 8: **OOD detection results w.r.t data augmentations on CIFAR100.**

| Score | Data Augmentaion | Near-OOD AUROC ↑ | Near-OOD FPR@95 ↓ | Far-OOD AUROC ↑ | Far-OOD FPR@95 ↓ | ID ACC ↑ |
|-------|------------------|-------|--------|-------|--------|--------|
| MLS | v1 | 81.05 | 55.47 | 79.67 | 56.73 | 77.25 |
|  | v1 + LS | 80.35 | 58.06 | 78.44 | 60.88 | 77.78 |
|  | v1 + mixup | 77.57 | 73.10 | 72.68 | 78.65 | 79.69 |
|  | v1 + RE | 80.96 | 56.48 | 82.31 | 52.20 | 76.91 |
|  | v1 + TA | 81.92 | 55.78 | 82.91 | 50.57 | 78.78 |
|  | v1+all augs (v2) | 78.34 | 71.64 | 74.85 | 71.46 | 79.89 |
| KNN | v1 | 80.18 | 61.23 | 82.40 | 53.65 | 77.26 |
|  | v1+LS | 78.84 | 61.34 | 81.24 | 56.14 | 77.78 |
|  | v1+mixup | 78.99 | 60.55 | 83.08 | 52.68 | 79.69 |
|  | v1+RE | 79.91 | 62.27 | 83.87 | 50.99 | 76.91 |
|  | v1+TA | 79.98 | 63.89 | 85.53 | 46.86 | 78.78 |
|  | v1+all augs (v2) | 79.69 | 60.98 | 85.09 | 48.70 | 79.89 |
| NNGuide | v1 | 80.27 | 58.36 | 81.41 | 56.66 | 77.26 |
|  | v1+LS | 41.33 | 91.19 | 43.48 | 90.77 | 77.78 |
|  | v1+mixup | 46.46 | 89.79 | 53.86 | 85.09 | 79.69 |
|  | v1+RE | 80.00 | 60.18 | 84.44 | 51.02 | 76.91 |
|  | v1+TA | 81.24 | 57.38 | 86.01 | 46.51 | 78.78 |
|  | v1+all augs (v2) | 37.00 | 92.39 | 49.74 | 85.67 | 79.89 |

Table 9: **OOD detection results w.r.t data augmentations on ImageNet200.**

| Score | Data Augmentaion | Near-OOD AUROC ↑ | Near-OOD FPR@95 ↓ | Far-OOD AUROC ↑ | Far-OOD FPR@95 ↓ | ID ACC ↑ |
|-------|------------------|-------|--------|-------|--------|--------|
| MLS | v1 | 82.90 | 59.76 | 91.11 | 34.03 | 86.37 |
|  | v1 +LS | 80.74 | 67.81 | 87.25 | 47.82 | 86.87 |
|  | v1 + mixup | 81.48 | 65.49 | 88.70 | 43.87 | 86.86 |
|  | v1 + RE | 82.57 | 60.40 | 90.94 | 34.52 | 86.54 |
|  | v1 + TA | 82.43 | 59.81 | 91.15 | 33.04 | 87.04 |
|  | v1+all augs (v2) | 79.18 | 72.55 | 86.50 | 52.88 | 86.85 |
| KNN | v1 | 81.59 | 58.26 | 91.49 | 31.15 | 86.37 |
|  | v1+LS | 81.37 | 58.46 | 91.14 | 31.49 | 86.87 |
|  | v1+mixup | 80.98 | 60.31 | 90.17 | 35.26 | 86.86 |
|  | v1+RE | 81.24 | 58.45 | 90.31 | 34.35 | 86.54 |
|  | v1+TA | 81.06 | 59.36 | 90.43 | 33.56 | 87.04 |
|  | v1+all augs (v2) | 81.24 | 58.49 | 89.70 | 36.64 | 86.85 |
| NNGuide | v1 | 82.54 | 63.10 | 93.11 | 30.70 | 86.37 |
|  | v1+LS | 66.65 | 79.65 | 76.31 | 63.78 | 86.87 |
|  | v1+mixup | 72.08 | 75.29 | 84.78 | 52.10 | 86.86 |
|  | v1+RE | 80.83 | 67.50 | 92.04 | 33.93 | 86.54 |
|  | v1+TA | 80.88 | 65.58 | 92.48 | 32.11 | 87.04 |
|  | v1+all augs (v2) | 44.72 | 88.82 | 55.15 | 81.17 | 86.85 |

## C.5   FINETUNE THE FULLY CONNECTED (FC) LAYER OR THE WHOLE NETWORK

AugDelete deletes all data augmentations and finetunes the FC layer while keeping features fixed, since label smoothing and mixup influence logits more than features. If further finetuning the feature extractor $G$, can we get additional gain in OOD detection? We finetune the entire network or FC layer for 15 epochs. Table 13 compares the results of finetuning FC or the whole network. Finetuning the entire network only gets marginal or no gains compared to finetuning FC layers. However, finetuning the whole network requires more time and delivers worse ID accuracy, because deleting all data augmentation during finetuning will hurt pretrained features.

## C.6   FIXING MIXUP FOR OOD DETECTION

Mixup is fixed in AugRevise with $L_{vs}$ loss to increase the separability between ID and mixed samples. Table 14 and 15 shows the quantitative results of fixing mixup. Regmixup improves the vanilla mixup but cannot outperform the v1 model in OOD detection. Adopting mixup in AugRevise can outperform the v1 model in both ID classification and OOD detection.

Table 10: **AugDelete for models trained with different data augmentations on CIFAR10.**

| Data Augmentaion | AugDelete | Near-OOD | | Far-OOD | | ID ACC |
| | | AUROC ↑ | FPR@95 ↓ | AUROC ↑ | FPR@95 ↓ | ↑ |
|---|---|---|---|---|---|---|
| v1 | ✗ | 87.52 | 61.32 | 91.10 | 41.67 | 95.06 |
| | ✓ | 88.01 | 56.84 | 91.40 | 37.56 | 95.01 |
| v1+LS | ✗ | 84.39 | 76.20 | 88.89 | 58.10 | 95.95 |
| | ✓ | 90.04 | 45.44 | 92.13 | 36.80 | 95.92 |
| v1+mixup | ✗ | 82.13 | 93.76 | 86.05 | 85.40 | 95.23 |
| | ✓ | 89.91 | 40.30 | 92.54 | 26.72 | 95.21 |
| v1+RE | ✗ | 88.47 | 56.22 | 91.99 | 39.53 | 95.36 |
| | ✓ | 89.13 | 50.47 | 92.48 | 34.87 | 95.42 |
| v1+TA | ✗ | 92.15 | 32.37 | 95.28 | 19.68 | 95.51 |
| | ✓ | 92.34 | 29.88 | 95.14 | 19.50 | 95.46 |
| v1+all augs (v2) | ✗ | 85.88 | 77.50 | 92.07 | 45.65 | 95.86 |
| | ✓ | 91.61 | 34.61 | 94.66 | 24.81 | 95.85 |

Table 11: **AugDelete for models trained with different data augmentations on CIFAR100.**

| Data Augmentaion | AugDelete | Near-OOD | | Far-OOD | | ID ACC |
| | | AUROC ↑ | FPR@95 ↓ | AUROC ↑ | FPR@95 ↓ | ↑ |
|---|---|---|---|---|---|---|
| v1 | ✗ | 81.05 | 55.47 | 79.67 | 56.73 | 77.25 |
| | ✓ | 80.96 | 55.60 | 80.26 | 56.01 | 77.16 |
| v1 + LS | ✗ | 80.35 | 58.06 | 78.44 | 60.88 | 77.78 |
| | ✓ | 80.81 | 55.89 | 79.67 | 58.17 | 77.93 |
| v1 + mixup | ✗ | 77.57 | 73.10 | 72.68 | 78.65 | 79.69 |
| | ✓ | 80.46 | 62.21 | 77.81 | 64.60 | 79.75 |
| v1 + RE | ✗ | 80.96 | 56.48 | 82.31 | 52.20 | 76.91 |
| | ✓ | 81.04 | 56.31 | 82.84 | 51.10 | 76.81 |
| v1 + TA | ✗ | 81.92 | 55.78 | 82.91 | 50.57 | 78.78 |
| | ✓ | 81.93 | 55.84 | 82.51 | 51.03 | 78.62 |
| v1+all augs (v2) | ✗ | 78.34 | 71.64 | 74.85 | 71.46 | 79.89 |
| | ✓ | 80.89 | 62.21 | 80.33 | 56.96 | 80.03 |

Table 12: **AugDelete for models trained with different data augmentations on ImageNet200.**

| Data Augmentaion | AugDelete | Near-OOD | | Far-OOD | | ID ACC |
| | | AUROC ↑ | FPR@95 ↓ | AUROC ↑ | FPR@95 ↓ | ↑ |
|---|---|---|---|---|---|---|
| v1 | ✗ | 82.90 | 59.76 | 91.11 | 34.04 | 86.37 |
| | ✓ | 82.92 | 59.24 | 90.51 | 35.82 | 86.26 |
| v1+LS | ✗ | 80.74 | 67.81 | 87.25 | 47.82 | 86.87 |
| | ✓ | 81.86 | 64.21 | 87.85 | 45.22 | 86.94 |
| v1+mixup | ✗ | 81.48 | 65.49 | 88.70 | 43.87 | 86.86 |
| | ✓ | 82.51 | 62.44 | 89.08 | 41.88 | 86.93 |
| v1+RE | ✗ | 82.57 | 60.40 | 90.94 | 34.52 | 86.54 |
| | ✓ | 83.06 | 59.43 | 90.69 | 35.49 | 86.63 |
| v1+TA | ✗ | 82.43 | 59.81 | 91.15 | 33.04 | 87.04 |
| | ✓ | 82.97 | 58.91 | 90.85 | 33.90 | 87.04 |
| v1+all augs (v2) | ✗ | 79.18 | 72.55 | 86.50 | 52.88 | 86.85 |
| | ✓ | 82.33 | 62.70 | 89.30 | 40.78 | 87.14 |

## C.7 COMPARE AUGDELETE AND AUGREVISE

We compare AugDelete and AugRevise in Table 16. AugRevise outperforms AugDelete and vanilla v1 models in both the ID classification and OOD detection. However, adding label smoothing in AugRevise will decrease the OOD performance in both near-OOD and far-OOD detection, suggesting that label smoothing should be removed in AugRevise.

Table 13: **Comparing finetuning the fully connected (FC) layer and finetuning the whole network.**

| Dataset | Models | Finetuning | Near-OOD | | Far-OOD | | ID ACC |
|---|---|---|---|---|---|---|---|
| | | | AUROC ↑ | FPR@95 ↓ | AUROC ↑ | FPR@95 ↓ | ↑ |
| ImageNet-1k | ResNet50-v2 | ✗ | 69.20 | 86.00 | 76.47 | 83.49 | 80.92 |
| | | FC | 78.69 | 70.05 | 87.47 | 56.18 | 80.31 |
| | | Network | 78.47 | 70.74 | 88.98 | 45.45 | 80.08 |
| ImageNet-1k | Swin-T-v2 | ✗ | 75.66 | 80.76 | 84.80 | 67.81 | 81.59 |
| | | FC | 81.01 | 69.06 | 90.96 | 37.79 | 81.30 |
| | | Network | 77.58 | 72.76 | 87.42 | 51.26 | 80.45 |

Table 14: **The results of fixing mixup on CIFAR10/100.** The top/bottom halves are for CIFAR10/CIFAR100, seperately.

| Data Augmentaion | Loss | Near-OOD | | Far-OOD | | ID ACC |
|---|---|---|---|---|---|---|
| | | AUROC ↑ | FPR@95 ↓ | AUROC ↑ | FPR@95 ↓ | ↑ |
| v1 | $L_{CE}$ | 87.52 | 61.32 | 91.10 | 41.68 | 95.06 |
| v1 + mixup | $L_{CE}^{mix}$ | 84.39 | 76.20 | 88.89 | 58.10 | 95.95 |
| v1 + regmixup | $L_{CE}^{rmix}$ | 89.19 | 53.81 | 93.18 | 32.93 | 96.27 |
| v1 + mixup-AugRevise | $L_{CE}^{rvmix}$ | 90.56 | 44.71 | 94.20 | 25.55 | 96.58 |
| v1 | $L_{CE}$ | 81.05 | 55.47 | 79.67 | 56.73 | 77.25 |
| v1 + mixup | $L_{CE}^{mix}$ | 77.57 | 73.10 | 72.68 | 78.65 | 79.69 |
| v1 + regmixup | $L_{CE}^{rmix}$ | 82.22 | 56.53 | 82.40 | 54.80 | 80.43 |
| v1 + mixup-AugRevise | $L_{CE}^{rvmix}$ | 83.34 | 51.56 | 85.20 | 45.93 | 81.22 |

Table 15: **The results of fixing mixup on ImageNet200.**

| Training Recipe | Loss | Near-OOD | | Far-OOD | | ID ACC |
|---|---|---|---|---|---|---|
| | | AUROC ↑ | FPR@95 ↓ | AUROC ↑ | FPR@95 ↓ | ↑ |
| v1 | $L_{CE}$ | 82.90 | 59.76 | 91.11 | 34.04 | 86.37 |
| v1+mixup | $L_{CE}^{mix}$ | 80.74 | 67.81 | 87.25 | 47.82 | 86.87 |
| v1+regmixup | $L_{CE}^{rmix}$ | 82.85 | 61.58 | 91.10 | 34.48 | 87.58 |
| v1+mixup-AugRevise | $L_{CE}^{rvmix}$ | 83.88 | 54.26 | 91.57 | 29.91 | 87.28 |

Table 16: **Compare AugDelete and AugRevise on CIFAR10/100 and ImageNet200.**

| ID Dataset | Train Recipe | Near-OOD | | Far-OOD | | ID ACC |
|---|---|---|---|---|---|---|
| | | AUROC ↑ | FPR@95 ↓ | AUROC ↑ | FPR@95 ↓ | ↑ |
| CIFAR10 | v1 | 87.52 | 61.32 | 91.10 | 41.67 | 95.06 |
| | v2 | 85.88 | 77.50 | 92.07 | 45.65 | 95.86 |
| | v2+AugDelete | 91.61 | 34.61 | 94.66 | 24.81 | 95.85 |
| | v2+AugRevise | 92.78 | 30.37 | 95.28 | 20.16 | 96.73 |
| | v2+AugRevise +LS | 91.34 | 39.72 | 93.97 | 31.34 | 96.78 |
| CIFAR100 | v1 | 81.05 | 55.46 | 79.67 | 56.72 | 77.26 |
| | v2 | 78.34 | 71.64 | 74.85 | 71.46 | 79.89 |
| | v2+AugDelete | 80.89 | 62.21 | 80.33 | 56.96 | 80.03 |
| | v2+AugRevise | 84.05 | 51.49 | 83.33 | 50.22 | 82.10 |
| | v2+AugRevise +LS | 83.83 | 50.87 | 82.21 | 51.17 | 81.85 |
| ImageNet200 | v1 | 82.90 | 59.76 | 91.11 | 34.04 | 86.37 |
| | v2 | 79.33 | 72.21 | 86.36 | 53.88 | 86.89 |
| | v2+AugDelete | 82.33 | 62.70 | 89.30 | 40.78 | 87.14 |
| | v2+AugRevise | 84.07 | 54.02 | 91.70 | 29.41 | 87.67 |
| | v2+AugRevise +LS | 83.88 | 54.64 | 90.47 | 33.10 | 87.33 |

## C.8 OOD DETECTION WITH VARIOUS PRETRAINED NETWORK ARCHITECTURES

We apply AugDelete to pretrained models with different network architectures including convolutional neural networks (CNNs) and transformers. Table 17 presents the ID accuracy and OOD performance

with/without AugDelete. We see that AugDelete improves the OOD detection of both CNNs and transformers while maintaining ID accuracy.

Table 17: **AugDelete w.r.t various pretrained Networks on ImageNet-1k.** The top half is before AugDelete while the bottom half is after AugDelete.

| Pre-trianed Models | Near-OOD | | Far-OOD | | ID ACC |
|---|---|---|---|---|---|
| | AUROC ↑ | FPR@95 ↓ | AUROC ↑ | FPR@95 ↓ | ↑ |
| ResNet50-v1 | 76.46 | 67.84 | 89.58 | 38.20 | 76.18 |
| ResNet50-v2 | 69.20 | 86.00 | 76.47 | 83.49 | 80.92 |
| MobileNetv2-v1 | 72.01 | 73.01 | 88.68 | 38.54 | 71.91 |
| MobileNetv2-v2 | 73.89 | 74.33 | 80.74 | 62.88 | 72.22 |
| ResNetXt50-v1 | 78.49 | 67.73 | 89.37 | 40.64 | 77.64 |
| ResNetXt50-v2 | 72.06 | 84.55 | 79.28 | 82.66 | 81.22 |
| WideResNet50-v1 | 78.69 | 67.93 | 89.02 | 41.22 | 78.50 |
| WideResNet50-v2 | 66.56 | 87.58 | 68.32 | 91.84 | 81.64 |
| RegNet-v1 | 78.58 | 70.92 | 88.13 | 45.16 | 80.44 |
| RegNet-v2 | 72.13 | 89.20 | 78.41 | 91.49 | 82.96 |
| ConvNext-v2 | 76.44 | 74.10 | 84.58 | 53.83 | 83.59 |
| Swin-T-v2 | 75.66 | 80.76 | 84.80 | 67.81 | 81.59 |
| ViT-B-16-v2 | 68.30 | 92.25 | 83.54 | 79.23 | 81.14 |
| ResNet50-v2 + AugDelete | 78.69 | 70.05 | 87.47 | 56.18 | 80.31 |
| MobileNetv2-v2 + AugDelete | 75.45 | 69.93 | 83.63 | 56.23 | 70.24 |
| ResNetXt50-v2 + AugDelete | 80.27 | 69.48 | 88.64 | 50.59 | 80.92 |
| WideResNet50-v2 + AugDelete | 76.67 | 77.19 | 83.84 | 71.38 | 81.45 |
| RegNet-v2 + AugDelete | 76.74 | 84.57 | 87.79 | 66.05 | 82.86 |
| ConvNext-v2 + AugDelete | 79.41 | 67.19 | 88.88 | 45.69 | 82.98 |
| Swin-T-v2 + AugDelete | 81.01 | 69.06 | 90.96 | 37.79 | 81.30 |
| ViT-B-16-v2 + AugDelete | 79.83 | 69.84 | 91.87 | 30.31 | 81.00 |

## C.9 COMPARISON WITH VARIOUS POST-HOC OOD DETECTION METHODS.

We combine AugDelete into various post-hoc OOD detection methods on Openood V1.5. Both logit-based and feature-and-logit-based methods are considered for AugDelete. Note that AugDelete has no effect on the feature-based method since it does not change the features. Table 18 shows the results with ResNet-v2, Swin-T-v2 and ViT-B-16 prerained networks. More results concerning network architectures are in the appendix. We can see that AugDelete can improve both methods by a large margin since AugDelete fixes the logits hurt by label smoothing and mixup.

## C.10 DETAILED RESULTS OF AUGREVISE FOR TRAINING-TIME MODEL ENHANCEMENT

We train models from scratch with AugRevise on ImageNet200/1k and CIFAR10/100 datasets, following the same training setting as OpenoodV1.5. ResNet18 is adopted for CIFAR10/100 and ImageNet200, while ResNet50 is for ImageNet200. Note that AugRevise for ImageNet-1k is trained 100 epochs as ResNet50-v1 instead of 600 epochs as ResNet50-v2. We choose logit-based, feature-based, and logit-and-feature-based OOD score functions for AugRevise. Table 19 and 20 compares AugRevise with state-of-the-art (SOTA) methods in Openood V1.5 Benchmark. AugRevise improves both logit-based and feature-based methods since it improves both features and logits. AugRevise also improves ID accuracy and outperforms comparing methods. Overall, AugRevise outperforms both post-hoc and training-based methods in ID and OOD.

Table 18: **AugDelete w.r.t various OOD scores on ImageNet-1k.**

| Pre-trianed Models | Method | AugDelete | Near-OOD AUROC ↑ | Near-OOD FPR@95 ↓ | Far-OOD AUROC ↑ | Far-OOD FPR@95 ↓ |
|---|---|---|---|---|---|---|
| ResNet50-v2 | MLS | ✗ | 69.20 | 86.00 | 76.47 | 83.49 |
| | | ✓ | 78.69 | 70.05 | 87.47 | 56.18 |
| | EBO | ✗ | 54.39 | 89.23 | 51.36 | 90.72 |
| | | ✓ | 76.84 | 72.46 | 86.82 | 58.72 |
| | MSP | ✗ | 72.53 | 82.21 | 81.48 | 72.28 |
| | | ✓ | 77.64 | 66.92 | 85.58 | 53.98 |
| | ASH | ✗ | 54.75 | 90.01 | 52.32 | 91.16 |
| | | ✓ | 76.50 | 72.32 | 86.90 | 57.91 |
| | KNN | ✗ | 70.76 | 73.48 | 89.07 | 36.71 |
| | MDS | ✗ | 76.62 | 69.17 | 93.74 | 26.88 |
| | NNGuide | ✗ | 61.16 | 82.25 | 70.37 | 61.89 |
| | | ✓ | 71.38 | 70.84 | 83.71 | 45.60 |
| Swin-T-v2 | MLS | ✗ | 75.66 | 80.76 | 84.80 | 67.81 |
| | | ✓ | 81.01 | 69.06 | 90.96 | 37.79 |
| | EBO | ✗ | 73.23 | 83.31 | 81.32 | 75.59 |
| | | ✓ | 80.78 | 71.74 | 91.40 | 38.32 |
| | MSP | ✗ | 76.75 | 71.06 | 86.30 | 49.16 |
| | | ✓ | 78.88 | 64.08 | 88.28 | 43.68 |
| | ASH | ✗ | 67.91 | 85.86 | 71.93 | 82.68 |
| | | ✓ | 78.46 | 76.89 | 89.30 | 45.75 |
| | KNN | ✗ | 71.62 | 71.76 | 89.37 | 34.12 |
| | MDS | ✗ | 75.18 | 68.65 | 91.49 | 29.87 |
| | NNGuide | ✗ | 67.92 | 84.99 | 85.36 | 50.77 |
| | | ✓ | 71.95 | 83.48 | 90.07 | 42.44 |
| ViT-B-16-v2 | MLS | ✗ | 68.30 | 92.25 | 83.54 | 79.23 |
| | | ✓ | 79.83 | 69.84 | 91.87 | 30.31 |
| | EBO | ✗ | 62.41 | 93.19 | 78.98 | 85.35 |
| | | ✓ | 80.13 | 70.90 | 92.69 | 27.94 |
| | MSP | ✗ | 73.52 | 81.85 | 86.04 | 51.69 |
| | | ✓ | 77.77 | 65.34 | 89.00 | 39.64 |
| | ASH | ✗ | 57.82 | 93.65 | 73.08 | 85.39 |
| | | ✓ | 79.71 | 71.99 | 93.01 | 27.96 |
| | KNN | ✗ | 74.11 | 70.47 | 90.81 | 31.93 |
| | MDS | ✗ | 79.04 | 66.12 | 92.60 | 29.97 |
| | NNGuide | ✗ | 60.40 | 89.89 | 81.74 | 59.86 |
| | | ✓ | 69.83 | 85.66 | 90.36 | 43.40 |

Table 19: **AugRevise w.r.t various OOD scores on CIFAR10/100 and ImageNet200/1k.**

| ID Dataset | Method | AugRevise | Near-OOD AUROC ↑ | Near-OOD FPR@95 ↓ | Far-OOD AUROC ↑ | Far-OOD FPR@95 ↓ | ID ACC ↑ |
|---|---|---|---|---|---|---|---|
| CIFAR10 | MLS | ✗ | 87.52 | 61.32 | 91.10 | 41.67 | 95.06 |
| | | ✓ | 92.78 | 30.37 | 95.28 | 20.16 | 96.73 |
| | EBO | ✗ | 87.58 | 61.32 | 91.21 | 41.70 | 95.06 |
| | | ✓ | 92.86 | 30.43 | 95.46 | 20.11 | 96.73 |
| | MSP | ✗ | 88.03 | 48.18 | 90.73 | 31.72 | 95.06 |
| | | ✓ | 92.44 | 28.62 | 94.56 | 19.59 | 96.73 |
| | ASH | ✗ | 87.54 | 61.23 | 91.13 | 41.60 | 95.06 |
| | | ✓ | 92.82 | 30.42 | 95.35 | 20.11 | 96.73 |
| | KNN | ✗ | 90.64 | 33.99 | 92.96 | 24.28 | 95.06 |
| | | ✓ | 93.69 | 27.46 | 96.22 | 16.20 | 96.73 |
| | NNGuide | ✗ | 83.54 | 78.57 | 86.95 | 65.86 | 95.06 |
| | | ✓ | 92.83 | 32.41 | 95.60 | 20.48 | 96.73 |
| CIFAR100 | MLS | ✗ | 81.05 | 55.46 | 79.67 | 56.72 | 77.26 |
| | | ✓ | 84.05 | 51.49 | 83.33 | 50.22 | 82.10 |
| | EBO | ✗ | 80.91 | 55.60 | 79.77 | 56.58 | 77.26 |
| | | ✓ | 83.88 | 51.66 | 84.29 | 49.59 | 82.10 |
| | MSP | ✗ | 80.27 | 54.79 | 77.76 | 58.70 | 77.26 |
| | | ✓ | 83.37 | 52.06 | 81.63 | 52.17 | 82.10 |
| | ASH | ✗ | 81.07 | 55.99 | 79.92 | 55.69 | 77.26 |
| | | ✓ | 83.78 | 52.15 | 84.51 | 49.19 | 82.10 |
| | KNN | ✗ | 80.18 | 61.23 | 82.40 | 53.65 | 77.26 |
| | | ✓ | 81.88 | 60.24 | 85.88 | 46.68 | 82.10 |
| | NNGuide | ✗ | 80.27 | 58.36 | 81.41 | 56.66 | 77.26 |
| | | ✓ | 83.92 | 52.89 | 85.10 | 46.14 | 82.10 |
| ImageNet200 | MLS | ✗ | 82.90 | 59.76 | 91.11 | 34.04 | 86.37 |
| | | ✓ | 84.07 | 54.02 | 91.70 | 29.41 | 87.67 |
| | EBO | ✗ | 82.50 | 60.22 | 90.86 | 34.86 | 86.37 |
| | | ✓ | 83.68 | 54.49 | 91.85 | 29.66 | 87.67 |
| | MSP | ✗ | 83.34 | 54.83 | 90.13 | 35.43 | 86.37 |
| | | ✓ | 84.09 | 53.91 | 91.69 | 29.38 | 87.67 |
| | ASH | ✗ | 82.76 | 59.82 | 91.63 | 32.68 | 86.37 |
| | | ✓ | 83.94 | 53.82 | 92.68 | 26.85 | 87.67 |
| | KNN | ✗ | 81.59 | 58.26 | 91.49 | 31.15 | 86.37 |
| | | ✓ | 81.34 | 56.45 | 92.65 | 26.88 | 87.67 |
| | NNGuide | ✗ | 82.54 | 63.10 | 93.11 | 30.70 | 86.37 |
| | | ✓ | 84.34 | 54.15 | 94.29 | 20.98 | 87.67 |
| ImageNet-1k | MLS | ✗ | 76.46 | 67.84 | 89.58 | 38.20 | 76.18 |
| | | ✓ | 79.23 | 62.37 | 90.30 | 35.74 | 77.70 |
| | EBO | ✗ | 75.89 | 68.56 | 89.47 | 38.40 | 76.18 |
| | | ✓ | 78.96 | 62.59 | 90.72 | 34.57 | 77.70 |
| | MSP | ✗ | 76.02 | 65.67 | 85.23 | 51.47 | 76.18 |
| | | ✓ | 79.25 | 62.36 | 90.31 | 35.75 | 77.70 |
| | ASH | ✗ | 76.41 | 66.85 | 91.52 | 32.39 | 76.18 |
| | | ✓ | 79.34 | 61.09 | 91.93 | 30.91 | 77.70 |
| | KNN | ✗ | 71.10 | 70.87 | 90.18 | 34.13 | 76.18 |
| | | ✓ | 72.60 | 69.82 | 92.52 | 28.67 | 77.70 |
| | NNGuide | ✗ | 78.80 | 63.89 | 94.56 | 25.73 | 76.18 |
| | | ✓ | 80.90 | 59.92 | 93.44 | 27.35 | 77.70 |

Table 20: **Comapre AugRevise with training-based OOD detection methods on CIFAR10/100 and ImageNet200/1k.**

| Dataset | Methods | Near-OOD | | Far-OOD | | ID ACC |
|---------|---------|----------|--------|---------|--------|--------|
| | | AUROC ↑ | FPR@95 ↓ | AUROC ↑ | FPR@95 ↓ | ↑ |
| CIFAR10 | T2FNorm+T2FNorm (Regmi et al., 2023) | 92.79 | 26.47 | 96.98 | 12.75 | 94.69 |
| | LogitNorm+MSP (Wei et al., 2022) | 92.33 | 29.34 | 96.74 | 13.81 | 94.30 |
| | VOS+EBO (Du et al., 2022) | 87.70 | 57.03 | 90.83 | 40.43 | 94.31 |
| | NPOS+KNN (Tao et al., 2023) | 89.78 | 32.64 | 94.07 | 20.59 | — |
| | CIDER+KNN (Ming et al., 2023) | 90.71 | 32.11 | 94.71 | 20.72 | — |
| | MOS+MOS (Huang & Li, 2021) | 71.45 | 78.72 | 76.41 | 62.90 | 94.83 |
| | AugMix+MSP (Hendrycks et al., 2020b) | 89.43 | 37.68 | 91.66 | 27.00 | 95.01 |
| | RegMixup+MSP (Pinto et al., 2022) | 87.47 | 48.78 | 90.25 | 36.30 | 95.75 |
| | AugRevise +MLS | 92.78 | 30.37 | 95.28 | 20.16 | 96.73 |
| | AugRevise +KNN | 93.69 | 27.46 | 96.22 | 16.20 | 96.73 |
| | AugRevise +NNGuide | 92.83 | 32.41 | 95.60 | 20.48 | 96.73 |
| CIFAR100 | T2FNorm+T2FNorm (Regmi et al., 2023) | 79.84 | 58.47 | 82.73 | 51.25 | 76.43 |
| | LogitNorm+MSP (Wei et al., 2022) | 78.47 | 62.89 | 81.53 | 53.61 | 76.34 |
| | VOS+EBO (Du et al., 2022) | 80.93 | 55.56 | 81.32 | 53.70 | 77.20 |
| | NPOS+KNN (Tao et al., 2023) | 78.35 | 63.35 | 82.29 | 51.13 | — |
| | CIDER+MSP (Ming et al., 2023) | 73.10 | 72.02 | 80.49 | 54.22 | — |
| | MOS+MOS (Huang & Li, 2021) | 80.40 | 56.05 | 80.17 | 57.28 | 76.98 |
| | AugMix+MSP (Hendrycks et al., 2020b) | 79.36 | 56.30 | 77.18 | 58.36 | 76.45 |
| | RegMixup+MSP (Pinto et al., 2022) | 80.83 | 56.12 | 79.04 | 57.50 | 79.32 |
| | AugRevise +MLS | 84.05 | 51.49 | 83.33 | 50.22 | 82.10 |
| | AugRevise +KNN | 81.88 | 60.24 | 85.88 | 46.68 | 82.10 |
| | AugRevise +NNGuide | 83.92 | 52.89 | 85.10 | 46.14 | 82.10 |
| ImageNet200 | T2FNorm+T2FNorm (Regmi et al., 2023) | 83.00 | 55.01 | 93.55 | 25.73 | 86.87 |
| | LogitNorm+MSP (Wei et al., 2022) | 82.66 | 54.46 | 93.04 | 26.11 | 86.04 |
| | VOS+EBO (Du et al., 2022) | 82.51 | 59.89 | 91.00 | 34.01 | 86.23 |
| | NPOS+KNN (Tao et al., 2023) | 79.40 | 62.09 | 94.49 | 21.76 | — |
| | CIDER+KNN (Ming et al., 2023) | 80.58 | 60.10 | 90.66 | 30.17 | — |
| | MOS+MOS (Huang & Li, 2021) | 69.84 | 71.60 | 80.46 | 51.56 | 85.60 |
| | AugMix+MSP (Hendrycks et al., 2020b) | 83.49 | 54.97 | 90.68 | 33.42 | 87.01 |
| | RegMixup+MSP (Pinto et al., 2022) | 84.13 | 68.92 | 90.81 | 30.31 | 87.25 |
| | AugRevise +MLS | 84.07 | 54.02 | 91.70 | 29.41 | 87.67 |
| | AugRevise +KNN | 81.34 | 56.45 | 92.65 | 26.88 | 87.67 |
| | AugRevise +NNGuide | 84.34 | 54.15 | 94.29 | 20.98 | 87.67 |
| ImageNet-1k | T2FNorm+T2FNorm (Regmi et al., 2023) | 73.08 | 69.14 | 91.92 | 31.24 | 76.76 |
| | LogitNorm+MSP (Wei et al., 2022) | 74.62 | 68.56 | 91.54 | 31.32 | 76.45 |
| | VOS+EBO (Du et al., 2022) | — | — | — | — | — |
| | NPOS+KNN (Tao et al., 2023) | — | — | — | — | — |
| | CIDER+KNN (Ming et al., 2023) | 68.97 | 71.69 | 92.18 | 28.69 | — |
| | MOS+MOS (Huang & Li, 2021) | 72.85 | 76.31 | 82.75 | 52.63 | 72.81 |
| | AugMix+MSP (Hendrycks et al., 2020b) | 77.49 | 64.45 | 86.67 | 46.94 | 77.63 |
| | RegMixup+MSP (Pinto et al., 2022) | 77.04 | 65.33 | 86.31 | 48.91 | 76.68 |
| | AugRevise +MLS | 79.23 | 62.37 | 90.30 | 35.74 | 77.70 |
| | AugRevise +KNN | 72.60 | 69.82 | 92.52 | 28.67 | 77.70 |
| | AugRevise +NNGuide | 80.90 | 59.92 | 93.44 | 27.35 | 77.70 |