# OpenReview forum: "Fixing Data Augmentations for Out-of-distribution Detection"
_ICLR.cc/2025/Conference — ICLR 2025 Conference Withdrawn Submission_

### Official Review · Reviewer_1b6P · 2024-10-31

**Soundness:** 2
**Presentation:** 2
**Contribution:** 2
**Rating:** 3
**Confidence:** 4

**Summary:**

This paper examines why mixup and label smoothing can enhance the performance of image classifiers but, unlike RandAugment, Style Augment, and AugMix, simultaneously lead to lower out-of-distribution (OOD) detection performance. It also proposes AugDelete and AugRevise—methods that maintain the classification performance of classifiers trained with label smoothing or mixup while improving their OOD detection performance. AugDelete is lightweight, as it fine-tunes only the penultimate layer of pre-trained classifiers, whereas AugRevise achieves even better performance than AugDelete. The authors validate their claims using ResNets as well as the CIFAR and ImageNet datasets.

**Strengths:**

1. The proposed method is lightweight and easy to implement.
2. The proposed method outperforms baselines in the tested settings.

**Weaknesses:**

1. There are many studies showing that models trained with self-supervised learning are effective for OOD detection [1,2,3,4]. Why must the classifier perform OOD detection simultaneously, rather than using experts for each task of classification and OOD detection?
2. Can this method be applied to models trained with self-supervised learning?
3. In the training recipe for achieving the best-performing classifier, are label smoothing or mixup essential components?
4. Does this research have significance from a transfer learning perspective?

[1] Tack et al., "CSI: Novelty Detection via Contrastive Learning on Distributionally Shifted Instances" NeurIPS 2020
[2] Ming et al., "Delving into out-of-distribution detection with vision-language representations" NeurIPS 2022
[3] Jiang et al., "Negative Label Guided OOD Detection with Pretrained Vision-Language Models" ICLR 2024
[4] Lee et al., "Textual Training for the Hassle-Free Removal of Unwanted Visual Data: Case Studies on OOD and Hateful Image Detection" NeurIPS 2024

**Questions:**

(Copied from Weaknesses)
1. There are many studies showing that models trained with self-supervised learning are effective for OOD detection. Why must the classifier perform OOD detection simultaneously, rather than using experts for each task of classification and OOD detection?
2. Can this method be applied to models trained with self-supervised learning?
3. In the training recipe for achieving the best-performing classifier, are label smoothing or mixup essential components?
4. Does this research have significance from a transfer learning perspective?

---

### Official Review · Reviewer_gTNV · 2024-10-31

**Soundness:** 2
**Presentation:** 2
**Contribution:** 2
**Rating:** 3
**Confidence:** 4

**Summary:**

This paper investigates the impact of data augmentation techniques on OOD detection, focusing primarily on Label Smoothing and Mixup. The authors find that while these methods improve in-distribution accuracy, they lead to a decline in OOD detection performance. The authors attribute this phenomenon to the fact that both Label Smoothing and Mixup decrease the maximal logits, with this reduction being more pronounced in ID data. To address this issue, the authors propose two methods to mitigate the performance degradation caused by Label Smoothing and Mixup.

**Strengths:**

The authors observe an interesting phenomenon: the torchvision v2 models perform poorly in OOD detection compared to the torchvision v1 models. They find that this is due to the improved training techniques used in the v2 models, such as Label Smoothing and Mixup, which reduces the the maximal logits and then reduces the OOD detection performance.

**Weaknesses:**

* The using Label Smoothing and Mixup reduces the maximal logit is obvious, I am more concerned with the authors' statement that “this reduction is more pronounced for in-distribution (ID) samples than for out-of-distribution (OOD) samples”. The authors try to prove this in Proposition 4.2. However, the authors make so many strong assumptions without stating why these assumptions hold, so that the logic of the proof is like "assume that A is correct, therefore A is correct" (lines 816 to 822). It would be beneficial if the authors could provide a clearer proof.

* From Table1 I observe that compared to v1 (trained with vanilla cross-entropy loss), the proposed v1+mixup-AugRevise and v1+LS-AugRevise only improve by 0.72 and 0.17, respectively, which is not exciting considering the additional computational cost and hyperparameters

* The proposed fixing method requires retraining, making the method less favorable. I think the contribution of this paper could be greatly enhanced if a post-hoc method could be used for fixing.

Typo: "Proposition 4.1" in line 255-256 should be "Proposition 4.2"

**Questions:**

see weakness

---

### Official Review · Reviewer_iowN · 2024-11-04

**Soundness:** 3
**Presentation:** 3
**Contribution:** 3
**Rating:** 6
**Confidence:** 3

**Summary:**

This paper addresses the impact of certain data augmentations on out-of-distribution (OOD) detection performance. The authors observe that two popular data augmentation techniques—label smoothing and mixup—although effective at improving in-distribution (ID) accuracy, degrade OOD detection performance. They provide empirical evidence and theoretical insights to explain this issue, highlighting that these techniques reduce the separability between ID and OOD samples in logit space, which is critical for effective OOD detection.

**Strengths:**

1. The motivation is clear and strong. The observation that using data-based data augmentation degrades the OOD detection of the model is new. The paper is to show its solutions.

2. The authors conduct extensive experiments across multiple architectures and benchmark datasets to support their claims.

3. This paper also provides theoretical analysis for the proposed approach.

**Weaknesses:**

1. The paper provides valuable insights into the effects of label-based augmentations (label smoothing and Mixup) on OOD detection. However, it would benefit from a broader exploration of other popular augmentation strategies, such as CutMix, to examine if these alternatives yield similar or contrasting impacts on OOD performance. Could you clarify the rationale behind selecting these specific four augmentation methods? It would be helpful to explain whether and how these choices align with the evolution of torchvision (from v1 to v2) and whether the findings could generalize to other augmentations.

**Questions:**

1. In Figure 2, it seems that "all augs (v2)" does not only reduce the OOD detection performance, but also reduce the ID accuracy. Please explain this apparent reduction in both OOD detection performance and ID accuracy.
In addition, could the authors consider grouping RE and TA together, and mixup and LS together, then adding these two new data points to Figure 2? This might provide additional insights into the combined effects of these augmentation strategies on both OOD detection and ID accuracy.

2. In Equations (3) and (7), the standard cross-entropy (CE) loss function is missing the "negative" sign. While optimization can still proceed with a positive formulation, it’s important to clarify this deviation from the standard notation to avoid confusion.

---

### Official Review · Reviewer_4bwp · 2024-11-04

**Soundness:** 2
**Presentation:** 2
**Contribution:** 3
**Rating:** 5
**Confidence:** 4

**Summary:**

The authors observe a drop in OOD detection performance of torchvision-v2 models compared to their v1 counterparts despite a gain in ID classification accuracy. They identify mixup and label smoothing as the root cause for the decrease in OOD detection performance, especially on logit-based detection methods via theoretical and experimental analysis. They devise two strategies to mitigate the problem: AugDelete finetunes the linear layer of pretrained models without the problematic augmentation strategies, and AugRevise adds a loss term regularizing the effect of the max-logit of samples with and without mixup for training from scratch. Experiments on the OpenOOD1.5 benchmark are provided.

**Strengths:**

- The paper is straightforward: the authors identify a problem (reduced OOD detection performance models from a certain training script), provide an explanation (mixup and label smoothing) and a fix for it
- The experiments identifying label smoothing and mixup as the problems are convincing and thorough
- Especially AugDelete, while being a simple method, shows consistent and believable improvements
- Investigations on the effects of training on OOD detection are often overlooked and a relevant subject to study

**Weaknesses:**

- While the paper tells a consistent and mostly believable story, its scope is somewhat limited. In particular, it focuses on models from the torchvision v2 hub that were trained from scratch on the respective datasets. How the findings translate to models from other codebases (e.g. timm) with more diverse pretraining settings (e.g. ImageNet21k, Laion, CLIP, distillation, …,) or zero-shot models is unclear. As recent studies [1,2] have shown, SOTA results are often achieved for bigger models with large pretraining schemes especially with feature-based methods, so those setups would be interesting to look at.
- It is unclear if there is additional computational cost associated with AugRevise and RegMixup. Since for those methods both the mixed and ‘clean’ sample are propagated through the network, this in principle doubles the batchsize and the computational cost (if the batch-size is fixed w.r.t. ‘Clean’ data, which is not explained in the paper). This would give AugRevise and RegMixup an unfair advantage over other baseline methods that only forward the mixed or only the clean samples.
- The claim that “feature-based methods are likely similarly compromised” is not backed by the provided experiments. For instance, the auroc differences in Table 7/8/9 between v1 and v2 for KNN are marginal, for CIFAR10 even the best-performing model is a v2 model with KNN. For AugRevise, additional feature-based methods like Mahalanobis distance, relative Mahalanobis Distance, Vim, … are omitted in the experiments
- AugRevise changes the from-scratch training compared to the torchvison v2 training script, but eventually still applies AugDelete, which sometimes leads to significantly lower ID accuracy (e.g. RN50 on IN-1k). It is unclear if other training methods, e.g. autoaugment or 3-Augment[4] or RSB [5] or others would not achieve similar results (potentially when combined with AugDelete). Also, to my understanding, only one model per dataset is investigated with AugRevise (ResNet-18 and ResNet-50).
- The authors claim that their “empirical results challenge the conventional understanding of ID and OOD performance correlation”, but similar observations have already been made in previous work, e.g. in [3]
- Proposition 4.2 relies on the assumption that the cosine similarity between ID samples is smaller than between ID and OOD samples. This is a somewhat strong assumption: If this were satisfied for most samples, it would allow to design of a good OOD detector based on cosine similarity. I would appreciate a discussion on the limitations of this assumption and how well it is justified.
- There are several issues regarding the presentation and the clarity of the paper (details below in Questions)

[1] Julian Bitterwolf, Maximilian Müller, and Matthias Hein. In or out? Fixing ImageNet out-of-
distribution detection evaluation. In ICML, 2023.

[2] Galil, I., Dabbah, M., and El-Yaniv, R. A framework for benchmarking class-out-of-distribution detection and its application to imagenet. In The Eleventh International Conference on Learning Representations, 2023

[3] Maximilian Müller, Matthias Hein. How to train your ViT for OOD detection, ICLR 2024 R2FM workshop

[4] Touvron, H., Cord, M., and Jegou, H. Deit iii: Revenge of the vit. ECCV, 2022.

[5] R. Wightman, H. Touvron, and H. Jégou, “ResNet strikes ack: An improved training procedure in timm,” arXiv preprint arXiv: 2110.00476, 2021

**Questions:**

- Could the authors clarify the computational cost of AugRevise and RegMixup?
- The v2 ResNet50 yields an ImageNet-1k accuracy of 80.92%. The accuracies in Table 4&5 for AugRevise are significantly lower and v2 models are omitted from the Table. Are there more setups where training with AugRevise leads to a significant drop in accuracy compared to the v2 models?


Regarding clarity:
- Section 5.2, especially lines 360-377 introduces the central part of AugRevise, but it is very short, which is in contrast to the previous Sections where the effects of Mixup/LS were explained thoroughly with several ablations. Section 5.2, in particular the introduction of the loss function requires, more explanation and justification
- Figures 1 and 2 are hard to read on paper. Larger markers and more distinguishable colours would help. It is not always clear which text belongs to which dot.
- Line 475: Should it be ImageNet-1k instead of ImageNet200?
- Throughout the paper (e.g. Figure 3 and 4, but also in most other places): Specifying for each Table and Figure which dataset (ID and OOD), which score and which model is reported would make the Tables and Figures more self-contained
- Regarding OpenOODv1.5 reporting: Which numbers are usually reported? In OpenOOD there is commonly a split between near and far OOD (as reported in the Appendix), but in the main paper, there is only one number. Is it the average?
- Throughout the paper: The larger Tables are hard to digest, as there are many numbers but little structure. I suggest making the best methods per model bold and grouping the same models, perhaps also adding row colours.
- In some Tables some methods are missing that appear in others, e.g. MDS in Table 19, ASH in Table 5 for AugRevise

---

### Note · Authors · 2024-11-13

I have read and agree with the venue's withdrawal policy on behalf of myself and my co-authors.